# Cas9 targeted enrichment of mobile elements using nanopore sequencing

Torrin L. McDonald[1,3], Weichen Zhou[2,3], Christopher P. Castro[2], Camille Mumm[1], Jessica A. Switzenberg [2], Ryan E. Mills [1,2✉] & Alan P. Boyle [1,2✉]

Mobile element insertions (MEIs) are repetitive genomic sequences that contribute to genetic variation and can lead to genetic disorders. Targeted and whole-genome approaches using short-read sequencing have been developed to identify reference and non-reference MEIs; however, the read length hampers detection of these elements in complex genomic regions. Here, we pair Cas9-targeted nanopore sequencing with computational methodologies to capture active MEIs in human genomes. We demonstrate parallel enrichment for distinct classes of MEIs, averaging 44% of reads on-targeted signals and exhibiting a 13.4-54x enrichment over whole-genome approaches. We show an individual flow cell can recover most MEIs (97% L1Hs, 93% *Alu*Yb, 51% *Alu*Ya, 99% SVA_F, and 65% SVA_E). We identify seventeen non-reference MEIs in GM12878 overlooked by modern, long-read analysis pipelines, primarily in repetitive genomic regions. This work introduces the utility of nanopore sequencing for MEI enrichment and lays the foundation for rapid discovery of elusive, repetitive genetic elements.

[1] Department of Human Genetics, University of Michigan, Ann Arbor, MI, USA. [2] Department of Computational Medicine and Bioinformatics, University of Michigan, Ann Arbor, MI, USA. [3]These authors contributed equally: Torrin L. McDonald, Weichen Zhou. ✉email: remills@umich.edu; apboyle@umich.edu

At least 45% of the human genome is composed of transposable element (TE)-derived sequences[1]. TEs can be subdivided into four major categories: (i) DNA transposons; (ii) long terminal repeat (LTR) retrotransposons; (iii) long interspersed elements (LINEs); and (iv) short interspersed elements (SINEs). L1 (or, LINE-1) represents a subclass of LINEs and L1-derived sequences comprise approximately 17% of the human genome[1,2]. *Alu* elements, a subclass of SINEs, are ancestrally derived from a dimerization of the 7SL RNA gene and make up 11% of the human genome, spread out over 1 million copies[3]. SVA (SINE-VNTR-*Alu*) elements are active chimeric elements that have recently evolved, and are derived from a SINE-R sequence coupled with a VNTR (variable number of tandem repeats) region and an *Alu*-like sequence[4]. An average human genome contains approximately 80–100 active full-length human-specific L1s (L1Hs)[5–7] and a small number of highly active, or "hot," L1Hs sequences, that are responsible for the bulk of human retrotransposition activity[5,6,8,9]. This includes the mobilization of *Alu*s and SVAs which require *trans*-acting factors from L1s to transpose[3]. Collectively, the result of such recent mobilization events are referred to as mobile element insertions (MEIs).

Regions harboring these repetitive elements have long been considered part of the "dark matter" of the genome with no expected impact on human phenotypes. However, recent studies indicate that at least some recent insertions indeed play a functional role in various aspects of the cell. L1-mediated retrotransposition events can be mutagenic, and germline retrotransposition events within the exons or introns of genes can result in null or hypomorphic expression alleles, leading to sporadic cases of human disease[10]. In addition, Lubelsky and Ulitsky demonstrated that sequences enriched in *Alu* repeats can drive nuclear localization of long RNAs in human cells[11]. An SVA element insertion was recently reported in an intron of TAF1 that ablated expression through aberrant splicing, and is a driving mutation in X-linked Dystonia-Parkinsonism[12]. Another study showed that a recurrent intronic deletion results in the exonization of an *Alu* element that is found in 6% of families with mild hemophilia A in France[13]. Somatic L1 retrotransposition can occur in neuronal progenitor cells[14–18], indicating a possible role for L1s in the etiology of neuropsychiatric diseases[19]. In addition, a mutagenic L1 insertion that disrupted the 16th exon of the APC gene has been shown to instigate colorectal tumor development[9]. Beyond a widespread repertoire of disease associations, mobile elements also influence large scale genome structure. Recent work has demonstrated that transposition events are associated with three dimensional genome organization, and the evolution of chromatin structure in human and mouse[20–22].

A tremendous effort has been made to understand the varied functional outcomes of active MEIs. Similar efforts are underway to capture and resolve MEIs to discover additional avenues of genetic pathogenesis[8,23–28]. While transformative, these studies were confounded by the shortcomings of existing sequencing methodologies and bioinformatics pipelines, and limited in their ability to access a large (~50%), highly repetitive proportion of the genome[29]. The difficulty in uniquely aligning short-read sequences to repetitive genomic regions likely leads to an under-representation of MEIs that have inserted within these regions. Several tools that have been developed to identify non-reference MEIs from whole genome short-read data, including the Mobile Element Locator Tool (MELT)[27], Mobster[30], Tangram[31], TEA[32], and others, are further restricted by the short read length and repetitive nature of mobile elements when resolving longer, non-reference insertions, such as L1Hs and SVA[29,33]. Experimental approaches from paired-end fosmid sequencing[8,34] to PCR capture-based approaches[23,25,35–38] have been developed to capture MEIs, but they have the disadvantage of being low throughput. Recent methods[26,29,39,40] combining short-read sequencing and MEI 3′ end capture techniques provide a more reliable way for the MEI discovery. Such approaches can be used in the investigation of single-cell MEI profiles[26,29], yet they too are hindered by the aforementioned disadvantages due to the short-read dependence.

The advent of long-read sequencing technologies provides a powerful tool for characterizing repeat-rich genomic regions by providing substantially longer sequence reads compared to traditional short-read platforms[41,42]. We have recently applied these technologies to demonstrate that there are at least 2-fold more polymorphic L1Hs sequences in human populations than previously thought[29]. Several existing tools and pipelines have the ability to resolve reference and non-reference MEIs in the human genomes; however, most require a whole-genome pipeline for haplotype assembly, local assembly, or cross-platform support[43–46]. This often necessitates whole genome long-read sequencing, which is currently cost-prohibitive at scale and precludes an in-depth exploration into the impact of MEIs on human biology and disease. One solution to these barriers is the application of Cas9 targeted sequence capture with long read sequencing that allows for alignment to unique flanking genomic regions[47]. This approach significantly lowers costs and enables a focused and efficient computational analysis for MEI discovery. Here, we demonstrate the utility of an in vitro Cas9 enrichment of targeted sequence elements combined with Oxford Nanopore long-read sequencing, and established computational methodologies to identify a set of MEIs (L1s, *Alu*Ys, and SVAs)[47] that account for over 80% of currently active mobile elements in the human genome[48–50]. This technology has been previously utilized to resolve a variety of genomic structural variants, including diseases associated with polynucleotide repeats and oncogenic translocation events[47,51–53]. By targeting the Cas9 to subfamily-specific sequences within each element and developing a computational pipeline (Nano-Pal) for analysis of Cas9-enriched nanopore sequencing data, we demonstrate enrichment of mobile elements across the genome that are both annotated and unannotated in the GRCh38 reference (reference and non-reference MEIs, respectively).

## Results

**Cas9 targeted enrichment strategy for mobile elements using nanopore sequencing.** We chose GM12878 (NA12878), a member of the CEPH pedigree number 1463 (GM12878, GM12891, and GM12892)[54], as the benchmark genome in this study. GM12878 is one of the most thoroughly investigated human genetic control samples and has been used in many large-scale genomic projects, such as HapMap[55], 1000 Genomes Project[34,56,57], the Human Genome Structural Variation Consortium[33,46], Genome In A Bottle[58,59], and reference genome improvement projects[60]. To precisely capture MEIs of interest, we applied Cas9 targeted nanopore sequencing[47,51–53] to enrich for five active subfamilies of MEIs (L1Hs, *Alu*Yb, *Alu*Ya, SVA_F, and SVA_E) in GM12878 (Fig. 1) as well as L1Hs in the corresponding parental samples.

We designed guide RNAs using unique subfamily-specific sequences within each element to maximize the specificity of Cas9 targeting (Fig. 2, see "Methods" section). Using this approach, we generated a list of candidate 20 bp guide RNAs for each MEI category (Fig. 2a–c). We selected an L1Hs candidate guide RNA with the "ACA" motif at 5929 bp of the L1Hs consensus sequence, which distinguishes the L1Hs subfamily from other L1 subfamilies (e.g., L1PA)[23,39,50]. For *Alu*Y and SVA, 18 unique guide RNA candidates were obtained (one for *Alu*Yb, three for *Alu*Ya, seven for SVA_F, and six for SVA_E) (see "Methods"

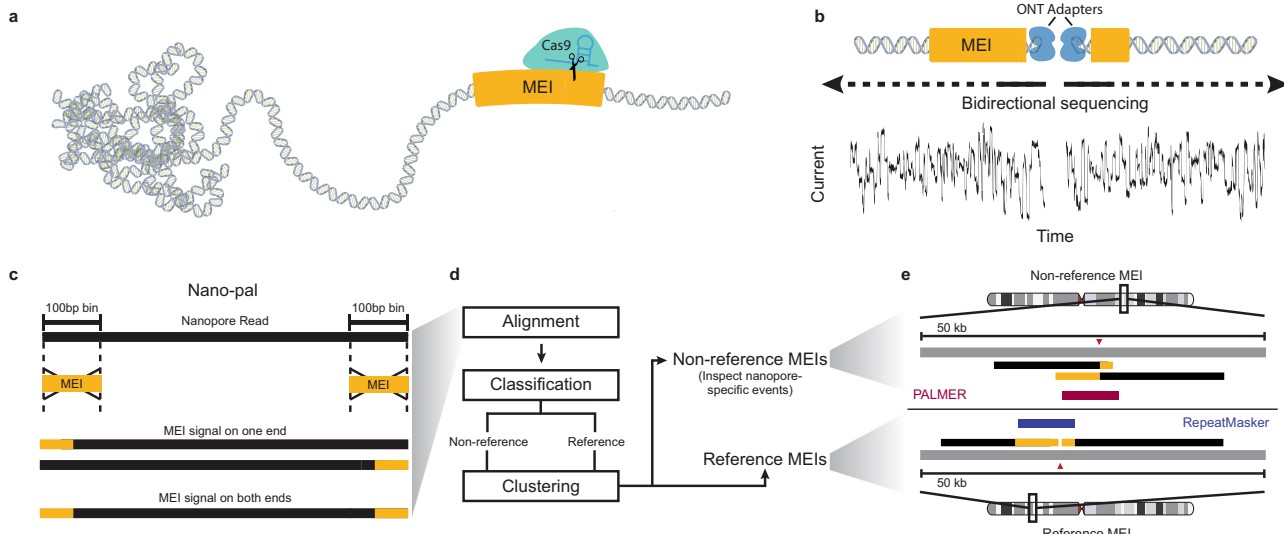

**Fig. 1 A schematic Cas9 targeted enrichment and Nano-Pal pipeline for mobile elements using nanopore sequencing. a** Purified genomic DNA (gDNA) is isolated by salting out and then extensively dephosphorylated. Dephosphorylated gDNA is incubated with the Cas9 ribonucleoprotein which is targeted to MEI subfamily-specific sequences near the 3′ end of the element. Taq polymerase (not shown), and dATPs (not shown) monoadenylate DNA ends. **b** Cas9 cleaved sites are ligated with Oxford Nanopore Technologies (ONT) sequencing adapters and sequenced on a flow cell. Sequencing is bi-directional from the cleavage site. **c** Nano-Pal scans the nanopore sequencing reads (black bars) after Cas9 enrichment for MEI signal on one or both ends. The yellow bar represents MEI consensus sequence or MEI signals in pairwise comparison of Nano-Pal. **d** All reads with or without annotated MEI signal are imported into the downstream pipeline. Alignment, classification, and clustering processes are sequentially conducted. Nano-Pal identifies reference and non-reference MEIs followed by the inspection of nanopore-specific non-reference MEIs (see "Methods" section). **e** Examples illustrating capture and alignment of reads containing non-reference L1Hs signal (top) and reference L1Hs signal (bottom). Aligned reads display a non-reference insertion (top) with L1Hs signal (yellow bar) and flanking genomic sequence (black bar). MEI components of reads in non-reference insertions are displayed as overlapping (soft clipping) due to lack of reference genome MEI annotation (gray bar). Aligned reads display annotated reference L1Hs (bottom, yellow bar), flanked by surrounding genomic sequence (black bar), separated by the Cas9 cleavage site (red triangle). PALMER and RepeatMasker tracks are illustrated in red and blue, respectively.

section, Supplementary Figs. 1 and 2 and Supplementary Data 1). Candidates were further prioritized to those with the largest number of subfamily-specific bases and proximity to the 3′ end of the MEI sequence, near the polyA tail, which is an obligate component of the TPRT mechanism of retrotransposition[61,62]. From the pool of candidates, a single guide RNA for each MEI subfamily was selected for downstream enrichment experiments (Supplementary Data 1). After Cas9 enrichment and read processing, we assessed the cleavage sites of all five guide RNAs (Fig. 2d and Supplementary Fig. 3). The resulting distribution showed a vast majority of the forward-strand reads start at the third or fourth base-distance from the "NGG" PAM site, and reverse-strand reads begin at the seventh base. This is consistent with previous characterization studies of Cas9 cleavage activity[63]. Furthermore, we observed strand bias with approximately 4.6-fold more reads on the forward strand compared to the reverse strand, which has been hypothesized to be caused by Cas9 remaining bound after cutting and obstructing adapter ligation and sequencing[47,64]. We detected directional sequencing biases within different MEI subfamilies and enrichment runs (Supplementary Data 2 and Supplementary Fig. 3).

We developed a computational pipeline, Nano-Pal, to analyze captured long reads after base-calling and trimming, estimate the on-target rate of Cas9 enrichment from MEI signals on the ends of reads, and identify reference and non-reference MEIs (Fig. 1b, c). Due to the frequency of targeted MEIs in the genome, an individual nanopore read may harbor a MEI signal on one or both ends. Reads with a single-end MEI signal had similar read lengths within all MEI experiments, yet were significantly larger than reads with two-end MEI signals. This was especially true in L1Hs experiments (L1Hs 1.9-fold, *Alu*Y 1.1-fold, SVA 1.4-fold,

Supplementary Fig. 4). To better distinguish non-reference MEI signals from those present in the reference, particularly where non-reference MEIs are embedded into reference MEIs[29], the pre-masking module from an enhanced version of our long-read non-reference MEI caller, PALMER[29], was implemented into Nano-Pal (Fig. 1d). This enabled identification and masking of reference MEIs in individual long-reads, enhancing detection of non-reference MEI signals within the remaining unmasked portion[29]. Non-reference and reference MEIs were then summarized by clustering nearby nanopore reads.

**Cas9 targeted enrichment efficiently captures mobile element signals in nanopore reads.** We performed a total of seven distinct MEI enrichment experiments across two types of nanopore sequencing platforms: Oxford Nanopore Flongles and MinIONs. Five of the experiments, one for each MEI subfamily, were sequenced on Flongle flow cells. One MinION flow cell was used to sequence an L1Hs enrichment, and another MinION flow cell was used to sequence a pooled sample of all five MEIs (see "Methods" section) (Table 1 and Supplementary Data 3). The N50 ranged from 14.9 to 32.3 kb in all the experiments after base calling and quality controls.

To estimate the enrichment efficiency for different flow cells and MEI subfamilies, all passed reads were classified into three categories: on-target, close-target, and off-target (see "Methods" section). The on-target rate for nanopore reads from a single L1Hs experiment on a Flongle flow cell, including both reference and non-reference MEIs, was 56.9%. Relatively lower on-target rates were observed for the other MEIs on the Flongle flow cell: 46.7% for *Alu*Yb, 23.8% for *Alu*Ya, 5.8% for SVA_F, and 2.3% for

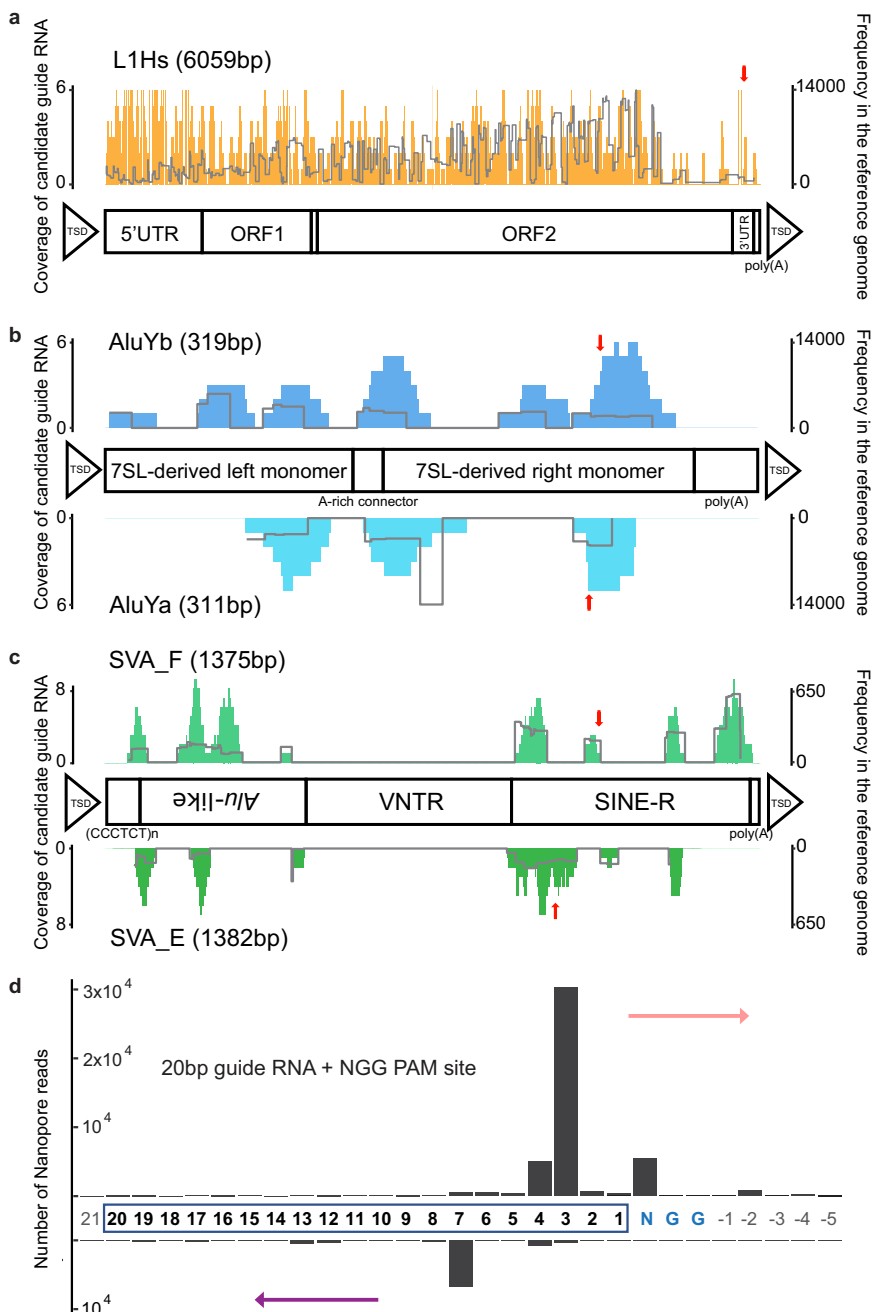

**Fig. 2 Guide RNA design for MEIs and guide RNA cleavage-site distribution. a** Distributions of candidate guide RNAs (left *Y*-axis and the histogram) in the L1Hs consensus sequence and structure information. The right *Y*-axis and the line indicate frequency of corresponding candidates in the reference genome sequence. **b** Upper panel shows the distribution for *Alu*Yb and the lower panel for *Alu*Ya. **c** Upper panel shows the distribution for SVA_F and the lower panel for SVA_E. Red arrows in **a**–**c** indicate where the selected guide is. **d** Cleavage-site distribution of all guide RNAs in this project. The *x*-axis indicates the position where the read ends or begins, with the number depicting the base distance from the PAM site (NGG). The PAM site (NGG) is colored blue and guide RNA bases are highlighted by a rectangle. Bases outside of the guide RNA or the PAM site are colored gray. The *y*-axis is the number of nanopore reads counted. The upper bar represents reads with forward strand sequencing outward from the 3′ end of the guide RNA (rose arrow), and the lower bar represents reads with reverse strand sequencing outward from the 5′ end of the guide RNA (purple arrow).

SVA_E (Table 1). When an L1Hs enrichment experiment was sequenced on a MinION flow cell, the on-target rate was approximately 35.0% (FAL11389) and 23.3% for a pooled MEI run (FAO84736). Compared to earlier studies (2.09% in Flongle and 4.61% in MinION)[47], these results show substantially improved enrichment, with a 1-fold to 25-fold increase relative to the Flongle flow cell, and over 5-fold enrichment relative to the MinION flow cell. These enrichment increases are likely due to the frequency of the targets in the genome. Overall, our approach

reaches an average of 44% of nanopore sequencing reads with target MEI signal from these seven flow cell runs.

To further assess the improved enrichment of MEI subfamilies, the extent of representation of MEI targets with high sequence identity were examined within the data. A portion of the enrichment data contained "close-target" reads that resemble related subfamilies of the intended targets, and can be explained by the base mismatch tolerance between the guide RNA sequence and the targeted MEI sequence[65,66]. For L1Hs, a rate of 16.8% on

**Table 1 Efficient enrichment of mobile element signals in nanopore reads.**

| MEI | Run | Flow cell | Read number | On-target | | Close-target | Off-target (%) |
|---|---|---|---|---|---|---|---|
| | | | | Reference (%) | Non-reference (%) | Reference (%) | |
| Individual | | | | | | | |
| L1Hs | ABB607 | Flongle | 4102 | 49.6 | 7.3 | 16.8 | 26.3 |
| *Alu*Yb | ACK645 | | 2271 | 40.2 | 6.5 | 1.0 | 52.2 |
| *Alu*Ya | ACK655 | | 12,513 | 18.0 | 5.8 | 10.3 | 65.9 |
| SVA_F | ACK629 | | 14,106 | 3.7 | 2.1 | 3.7 | 90.6 |
| SVA_E | ABO395 | | 7297 | 1.7 | 0.6 | 0.2 | 97.5 |
| L1Hs | FAL11389 | MinION | 110,029 | 30.7 | 4.3 | 37.9 | 27.1 |
| Pooled | FAO84736 | MinION | 105,410 | 20.1 | 3.2 | 38.4 | 38.3 |
| L1Hs | | | | 8.9 | 1.6 | 33.9 | |
| *Alu*Yb | | | | 7.0 | 1.2 | 3.2 | |
| *Alu*Ya | | | | 2.8 | | | |
| SVA_F | | | | 1.2 | 0.4 | 1.3 | |
| SVA_E | | | | 0.2 | | | |

Summary of seven representative flow cells: five individual Flongle flow cells for L1Hs (ABB607), *Alu*Yb (ACK645), *Alu*Ya (ACK655), SVA_F (ACK629), and SVA_E (ABO395) each, one individual MinION flow cell for L1Hs (FAL11389), and one pooled MinION flow cell for five MEIs (FAO84736).

the Flongle and 33.9–37.9% on the MinION flow cell was observed, with close-target reads mapping to reference L1PA regions. Flongle sequencing of *Alu*Ya had a rate of 10.3% of close-target reads to other reference *Alu*Y elements, in contrast to *Alu*Yb where a dramatically reduced "close-target" rate of 1.0% was observed (Table 1). This enhanced specificity may be explained by a specific insertion sequence within *Alu*Yb (5′-CAGTCCG-3′) that was included in the guide RNA, and is unique to the youngest *Alu* elements (*Alu*Yb)[49] of the genome. For the SVA Flongle sequencing, "close-target" rates of 3.7 and 0.2% to the other reference SVAs were observed in the SVA_F and SVA_E enrichment, respectively.

We next assessed the efficiency of our target enrichment of MEIs compared to whole-genome sequencing (WGS) approaches. A recent, related study used a whole-genome nanopore sequencing approach[45] to study MEIs and methylation, and provides an excellent benchmark to which we may compare our results. When taking total sequenced reads into account, our targeted approach exhibited between a 13.4–54-fold increase in the average number of reads per MEI compared to WGS (Supplementary Fig. 5). Furthermore, our read length N50 ranged from 14.9 to 32.3 Kbp compared to 5.14–10.57 Kbp reported in Ewing et al., suggesting that our targeted approach also results in a higher number of MEI spanning reads. Overall, these comparisons indicate that on the basis of per-base sequenced, MEI target capture exhibits significant enrichment advantages over whole genome approaches.

**Cas9 enrichment and nanopore sequencing rapidly saturates reference and non-reference MEIs.** Due to the possibility that the guide RNA may bind to off-target sites and direct Cas9 to cut in MEIs that are not perfectly matched to the guide sequence[65,66], we established expectation thresholds to evaluate the number of captured reference and non-reference MEIs. The reference MEI sets were obtained from RepeatMasker[67]. A "PacBio-MEI" callset in GM12878 was generated by comprising a mapping-based callset, and an assembly-based callset as a comprehensive gold standard set for non-reference MEIs (see "Methods" section). The PacBio-MEI includes 215 L1Hs, 362 *Alu*Yb and 593 *Alu*Ya in 1404 *Alu*s, and 33 SVA_F and 24 SVA_E in 72 SVAs (Supplementary Fig. 6 and Supplementary Data 4). Three categories (lower, intermediate, and upper) were defined that contain a number of reference and non-reference MEIs from each subfamily (Supplementary Data 5). Each threshold classifies MEIs

depending on the extent of allowed mismatches between the guide and the sequence. The lower-bound is the most stringent and requires a perfect match between the guide sequence and the MEI. The intermediate-bound is less stringent and can tolerate three or fewer mismatches. We consider this to be the closest estimation to the number of MEIs that a guide RNA could reasonably capture among these three boundaries. Finally, the upper-bound is the least stringent and most inflated, requiring that at least 60% of the guide sequence matches the MEI (see "Methods" section).

Upon comparing our MEI enrichment data to the aforementioned intermediate value estimates (Fig. 3a, b, Supplementary Fig. 7, and Supplementary Data 5 and 6), the individual and the pooled MinION flow cell captured 100% (35.8 mean coverage) and 96.0% (10.5 mean coverage) of known reference L1Hs with on-target reads, respectively. The individual Flongle captured 81.2% (2.7 reads mean coverage) of known reference L1Hs. For the non-reference L1Hs, the individual and pooled MinION flow cell captured 100% and 99.4% for all the L1Hs subfamilies, respectively. The individual Flongle captured 66.4%, 83.3%, and 95.2% for non-reference L1Ta, L1PreTa, and L1Ambig, respectively. Our results showed that only one of the MinION flow cells (FAL11389 or FAO84736) was necessary to capture most of the known reference and estimated non-reference L1Hs subfamilies in the genome when considering intermediate values, indicating a very high sensitivity of guide RNA targeting in the experiments.

Compared to the least stringent upper-bound estimates, 64.5% and 79.7% of known reference L1Hs were captured using the individual Flongle and MinION flow cell, respectively. Non-reference L1Hs capture ranged from 53.0 to 80.0%, compared to 4.1–7.3% of close-target L1PA elements, and less than 0.01% of off-targeting to other L1 elements (Fig. 3a, b, Supplementary Fig. 7, and Supplementary Data 5 and 6). The high percentage of elements captured that were on-target versus the other categories, including off-target, indicates the high specificity of the guide RNA to L1Hs in the enrichment. The read depth of the reference and non-reference L1Hs elements observed in these MinION flow cells has an approximate ratio of 2:1 (Fig. 3b), consistent with the expectation that reference MEIs are homozygous, and a considerable portion of non-reference MEIs are heterozygous[68].

For *Alu*Y subfamilies, individual Flongles were utilized for separate runs of *Alu*Yb (ACK645) and *Alu*Ya (ACK655), and one pooled MinION flow cell (FAO84736) (Fig. 3c, d, Supplementary Fig. 7, and Supplementary Data 5 and 6). 93.9% (3.5 mean

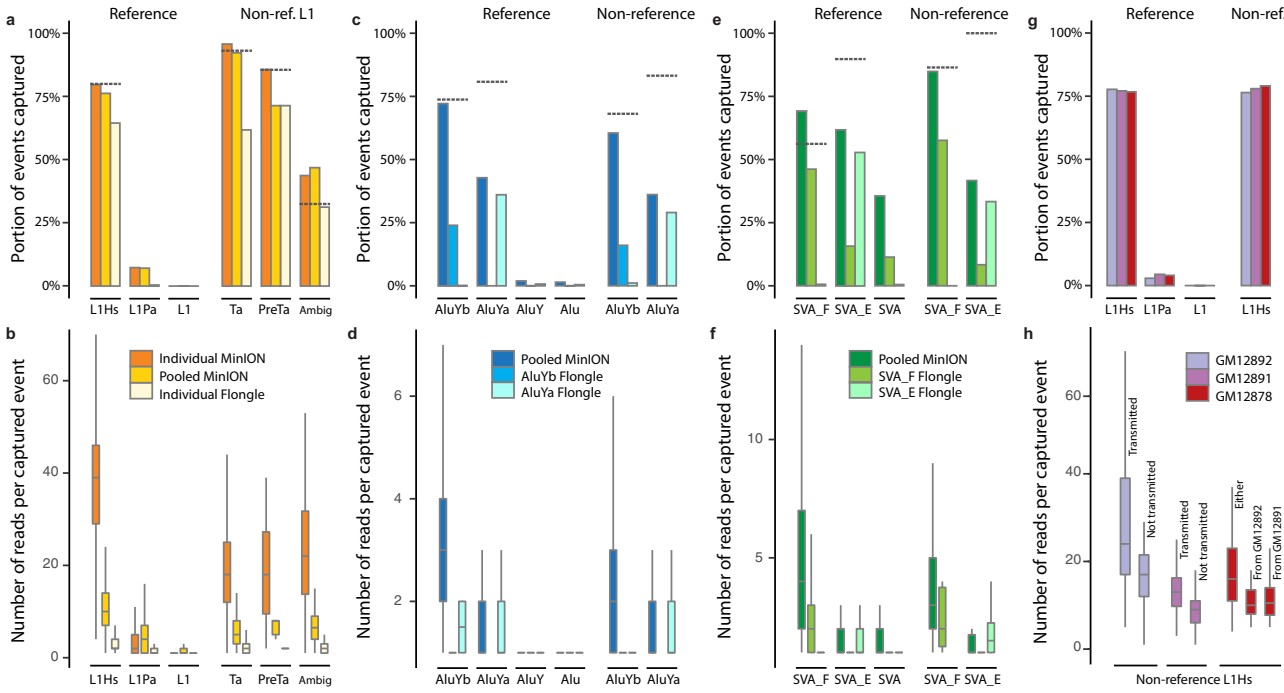

**Fig. 3 Systematic evaluation of known MEIs captured by nanopore Cas9 enrichment approach in different flow cells. a** Known L1Hs in GM12878 recovered by Cas9 targeted enrichment from the individual MinION flow cell (FAL11389), pooled-MEI MinION flow cell (FAO84736), and individual Flongle flow cell (ABB607), displayed as a proportion of the upper-bound known reference L1Hs, L1Pa, and other L1 as well as non-reference (non-ref.) L1Hs from the PacBio-MEI set. Non-reference L1Hs were divided into different subfamilies (L1Ta, L1PreTa, and L1Hs with ambiguous subfamilies). Dotted-gray line represents the intermediate values (as proportion) of MEIs that the guide RNA binds when allowing a ≤ 3 bp mismatch or gap. **b** Number of supporting reads of each captured L1 in the context of **a**. **c** Known *Alu*Y elements in GM12878 recovered by Cas9 enrichment on one pooled MinION flow cell (FAO84736), one individual *Alu*Yb Flongle flow cell (ACK645), and one individual *Alu*Ya Flongle flow cell (ACK655). **d** The number of supporting reads of each captured *Alu* element in the context of **c**. **e** Known SVA elements in GM12878 recovered by Cas9 enrichment on one pooled MinION flow cell (FAO84736), one individual SVA_F Flongle flow cell (ACK629), and one individual SVA_E Flongle flow cell (ACK395). **f** The number of supporting reads of each captured SVA element in the context of **e**. **g** Known L1Hs captured in the GM12878 trio by Cas9 enrichment on one pooled MinION flow cell (FAL15177). **h** The number of supporting reads of each captured non-reference L1Hs based on transmission in the GM12878 trio. The non-reference L1Hs in the parents (GM12892 and GM12891) were categorized into transmitted and not-transmitted. The non-reference L1Hs in the child (GM12878) were categorized as insertions inherited from GM12892 or GM12891, and from either parents (unknown parental lineage). In **b**, **d**, **f**, **h**, the numbers of captured MEI subfamily can be found in Supplementary Data 6 with information of mean and standard deviation; The error bars of boxplot range from $Q1 - 1.5$ IQR to $Q3 + 1.5$ IQR (IQR, interquartile range) and outliers are not shown.

coverage) reference and 88.0% (2.4 mean coverage) non-reference *Alu*Ybs were captured from the pooled MinION flow cell run, based on intermediate values. Similar to the L1Hs, the MinION flow cell was able to capture a vast majority of *Alu*Yb elements when considering intermediate values, indicating high sensitivity performance of the *Alu*Yb enrichment. A relatively lower rate of capture for reference (52.7%, 1.6 mean coverage) and non-reference (42.0%, 1.4 mean coverage) *Alu*Ya enrichment was observed in the pooled MinION flow cell run based on intermediate values. Cross-capture rate from the two individual Flongle flow cells was less than 0.1%, and close-target and off-target reference elements were <0.1 and 2% for the Flongles and MinION, respectively, indicating a high specificity of the guide RNA for each *Alu*Y subfamily (Fig. 3c and Supplementary Data 5 and 6).

Similar enrichment rates were obtained from the two individual Flongle flow cells for SVA_F (ACK629) and SVA_E (ACK395), and in the pooled MinION flow cell (FAO84736) (Fig. 3e, f, Supplementary Fig. 7, and Supplementary Data 5 and 6). 100% (4.6 mean coverage) reference and 96.6% (3.9 mean coverage) non-reference SVA_F were captured from the pooled MinION flow cell run based on intermediate values. A relatively lower rate of capture for reference (68.7%, 1.9 mean coverage) and non-reference (41.7%, 1.5 mean coverage) SVA_E enrichment was observed in the pooled MinION flow cell run based on

intermediate values. The close-target reference SVAs capture rate was relatively high in two of the runs (35.6% in MinION flow cell and 11.4% in ACK629 Flongle flow cell). This could be due to SVAs sharing less base pair substitutions among their subfamilies compared to the other MEI families, as it is the youngest retrotransposon family found in the hominid lineage[48].

Our results indicate that an individual MinION flow cell (FAL11389) is able to completely (100%) capture reference and non-reference instances of a single MEI subfamily (L1Hs) compared to sequencing on the smaller Flongle flow cells. More importantly, a pooled run of an unbarcoded, five MEI subfamily enrichment experiment can recover the vast majority of known reference and non-reference MEIs (96.5% L1Hs, 93.3% *Alu*Yb, 51.4% *Alu*Ya, 99.6% SVA_F, and 64.5% SVA_E) in the genome when considering elements with a ≤ 3 bp mismatch to the guide RNA. Such an approach outperformed individual Flongle flow cells and approached the same capture level as the single MEI subfamily MinION run (Fig. 3, Supplementary Fig. 7, and Supplementary Data 3, 5, and 6). This suggests that the MinION flow cell has ample sequencing capacity to accommodate each experiment with negligible competition between samples, despite sequencing multiple MEI enrichments on one platform. Finally, a substantial fraction of reference and non-reference MEI events can be captured in a single MinION sequencing experiment with

multiple supporting reads. With further optimization of the enrichment and sequencing methodologies, it is plausible to fully saturate reference and non-reference MEIs in a single experimental iteration.

**Detectable transmission of non-reference L1Hs within a trio.** To trace the transmission of non-reference L1Hs in GM12878 from the parents, another enrichment experiment of L1Hs elements was performed in GM12878, GM12891, and GM12892 (Fig. 3g, h and Supplementary Data 3, 6). The on-target rate for L1Hs ranged from 34.3 to 40.4% in the individual Flongles for GM12891 and GM12892 (parents), which mirrors the on-target rate of GM12878 (child) (Supplementary Data 7).

Transmission of non-reference L1Hs to GM12878 from the parental genomes was further examined using available GM12891 and GM12892 sequencing data. As the parental genomes lack sufficient long-read sequencing data, we utilized MELT to resolve non-reference L1Hs callsets from high-coverage Illumina short-read sequencing data. This analysis yielded 123 and 118 high confidence, non-reference L1Hs in GM12892 and GM12891, respectively. The number of MEIs identified in the MELT call sets are relatively lower than the number ($n = 205$) detected in the "PacBio-MEI" set for GM12878, consistent with previous observations that long-reads are more sensitive for MEI discovery[29,33,46] (Supplementary Data 4). Additional evidence of transmission can be derived by the enrichment of reads from non-reference MEIs in GM12878. We expect MEIs that can be transmitted from either parent will have higher read coverage due to a portion of these being homozygous, and single parent transmitted MEIs will be heterozygous in GM12878. As predicted, an enrichment of approximately 1.52-fold (17.6 vs. 11.6 mean coverage) was observed for these reads (Fig. 3h and Supplementary Data 6). Similarly, the "not transmitted to child" non-reference L1Hs in parent samples should be heterozygous and were observed to be depleted by approximately 0.56–0.72-fold (17.4 vs. 31.3 mean coverage in GM12892 and 10.7 vs. 14.8 mean coverage in GM12891) of the nanopore reads that have been transmitted to the child (Fig. 3h). These observations showed an expected supporting-read distribution of non-reference L1Hs, supporting the efficient nanopore Cas9 targeted enrichment in the pooled trio samples.

**Cas9 enrichment and nanopore sequencing captures non-reference mobile elements in complex genomic regions.** To estimate the efficacy of enrichment for non-reference MEIs with different sequencing coverage, we manually inspected each non-reference MEI reported by Nano-Pal and performed subsequent saturation analysis for all flow cells (Fig. 4, see "Methods" section). We find few additional L1Hs insertions by including additional on-target reads beyond approximately 30,000, using a cutoff of 15 supporting reads (Fig. 4a). This is consistent with the observation that the MinION (individual or pooled, usually with >100 k passed reads) has the ability to capture most non-reference L1Hs. In addition, there was no observable enrichment bias of MEI subfamilies from different flow cells (Supplementary Fig. 8).

We examined the 182 non-reference L1Hs in GM12878 that overlapped with the PacBio-MEI set. Of these 175 (96.2%) could be accounted for by the parental (GM12891 and GM12892) sequencing data (Supplementary Fig. 9), and three overlapped known polymorphic insertions[46]. The remaining four non-reference L1Hs are located within centromeric regions, which could be missed in the parental samples due to lack of supporting reads. In addition, we observed 601 non-reference *Alu*Y (including 323 *Alu*Yb and 263 *Alu*Ya) and 49 non-reference SVA (including 30 SVA_F and 15 SVA_E) that overlapped with

the PacBio-MEI set. We further examined the set of MEIs that were captured exclusively by Cas9 targeted enrichment and nanopore sequencing, but not found in the PacBio-MEI intersection (Supplementary Data 8). We identified 12 additional L1Hs insertions as nanopore specific with ≥4 supporting reads that had been missed by the PacBio-MEI set with valid hallmarks, including target site duplication motifs, poly(A), EN Cleavage site, and empty site sequences, indicating a retrotransposition event induced by target-primed reverse transcription mechanism (TRPT) (Supplementary Data 9 and Supplementary Fig. 10). In addition, we detected five *Alu*Y elements that were specifically captured by nanopore reads in the GM12878 genome (Supplementary Data 9). After refinement and inspection, we generated a full set of non-reference MEIs (194 L1Hs, 606 *Alu*Y, and 49 SVA) captured by Cas9 enrichment and nanopore sequencing in the GM12878 genome (Supplementary Data 10). Of note, all intermediate-value calls of PacBio-MEI for L1Hs, *Alu*Yb, and SVA_F were recovered in this study (Supplementary Data 5, 9, and 10). Additionally, 46 calls with L1Hs sequence and 14 with *Alu*Y sequence were captured by Cas9 target nanopore sequencing, yet not included in the final non-reference callset (Supplementary Data 8). Though lacking the support of TPRT hallmarks indicating a retrotransposition event, they may represent polymorphic duplicated sequences harboring an existing L1Hs or *Alu*Y element.

One non-reference L1Hs insertion at chrX:121,709,076 was particularly intriguing. The PacBio genome assembly-based approach overlooked this insertion, as it fell within a "reference L1 rich" region (Fig. 4d). Upon further inspection, this event was supported as a 403 bp heterozygous L1Hs insertion by the existence of significant retrotransposition hallmarks, as well as recurrence (dot) plots[29,69] (Fig. 4e, see "Methods" section). This insertion also shares a high sequence identity with a nearby reference L1PA11 element (Supplementary Data 8). The decrease in efficacy of the PacBio assembly-based approach in this region could be explained by the intricate nested "L1 in L1" structure and observed heterozygosity (Fig. 4d). Likewise, the *Alu*Y nanopore Cas9 enrichments captured interesting non-reference *Alu*Y instances: A homozygous *Alu*Yb8 insertion at chr19:52384635, an exonic region within the *ZNF880* gene, was reported to alter RNA expression due to the *Alu* element's effects on the RNA secondary structure[70]. Another heterozygous *Alu*Ya5 insertion at chr16:69157709 was located within a reference *Alu*Jr region, indicating a potential nested "*Alu* in *Alu*" structure that could hinder non-reference *Alu*Y discovery. These observations demonstrate the high sensitivity of nanopore Cas9 enrichment, suggesting its feasibility for MEI discovery in complex genomic regions.

## Discussion
Here, we describe our design and implementation of Cas9 targeted nanopore sequencing to enrich for retrotransposition competent, repetitive mobile elements in the human genome[47]. After carefully designing guide RNAs to each MEI subfamily and coupling the enrichment with an established computational pipeline, our approach reaches an average of 44% nanopore sequencing reads with target MEI signals. We recovered a vast majority of reference and known non-reference mobile elements (96.5% L1Hs, 93.3% *Alu*Yb, 51.4% *Alu*Ya, 99.6% SVA_F, and 64.5% SVA_E) in the genome using only a single MinION flow cell. In addition, we discovered 21 non-reference MEIs within the GM12878 genome that were previously missed by other orthogonal long-read pipelines. Our data suggest that a MinION flow cell is ideal for a pooled, multiple-element enrichment experiment, as a prohibitively reduced enrichment or extensive

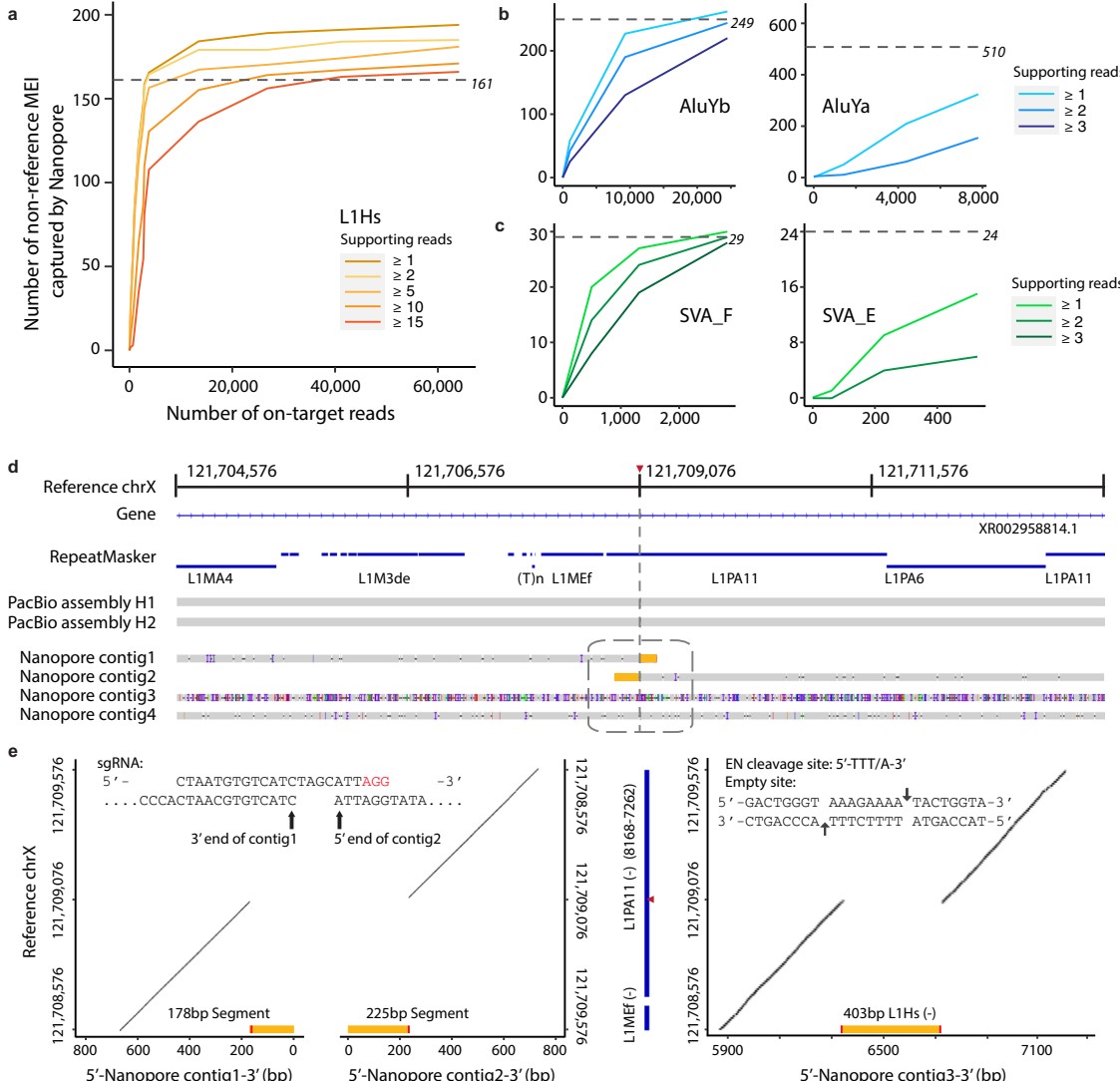

**Fig. 4 Non-reference MEIs captured by nanopore Cas9 enrichment approach. a** Number of non-reference L1Hs captured by nanopore Cas9 enrichment at different on-target read coverages for different supporting read cutoffs. The dotted-gray line with italic number represents the theoretical number of MEIs that the guide RNA binds when allowing a ≤ 3 bp mismatch or gap in the PacBio-MEI set. **b, c** Number of non-reference *Alu*Yb, *Alu*Ya, SVA_F, and SVA_E, respectively, captured by nanopore Cas9 enrichment at different on-target read coverages. Axis labels and theoretic guide number as in **a**. **d** An example of non-reference L1Hs specifically captured by nanopore sequencing at chrX:121,709,076. The tracks from top to bottom are as follows: reference coordinates with a red triangle represent the insertion site, gene track, RepeatMasker track (blue bars) with reference element annotation, PacBio contigs assembly for two haplotypes, four nanopore local-assembled contigs by CANU from different classifications of nanopore reads based on insertion signals (contig1, signal on 3′ end; contig2, signal on 5′ end; contig3, signal in the middle of the read; and contig4, no signal). **e** Recurrence (dot) plots for nanopore contigs versus the reference region chrX:121,708,576-121,7089,576 sequence. Left panel shows the most 3′ end of contig1 and the most 5′ end of contig2 versus the reference sequence. Yellow bar represents the non-reference L1Hs sequence contained in the contig. The red bar represents one side of the target site duplication motif for the non-reference L1Hs contained in the contig. The upper part of this panel demonstrates sequences at the end of two contigs regarding the cleavage site when aligning to the guide RNA sequence. Blue bars in the middle panel represent the RepeatMasker track with reference L1 information annotated, and the red triangle represents the insertion site in the reference L1 region. The right panel shows contig3 versus the reference sequence. Details of this non-reference L1Hs are detailed in the panel, including length, strand, empty site, and endonuclease (EN) cleavage site sequence.

cross-capturing of subfamilies was not observed. However, we observe that some of our MEIs targets (SVAs and *Alu*s) have reduced enrichment compared to L1Hs. While we find relatively stable on-target rates for a specific guide target, differences in guide target rates can be expected due to different numbers of genomic integration sites, high numbers of similar target sequences, or guide RNA efficiency variation. For example, the low number of SVAs in the genome resulted in rather low on-target reads, but sufficient coverage to identify most of these elements in the genome. Similarly, the high number of *Alu*s in the genome increases the set of near-matched guides and so we

have a high background of other *Alu*s enriched. The work presented here highlights the potential of targeted enrichment and nanopore sequencing to rapidly discover distinct MEIs, and cements an experimental foundation to probe even the most elusive mobile element insertion events.

Cas9 targeted enrichment paired with nanopore sequencing has the potential for resolving complex structural variation, previously obfuscated by sequencing and computational limitations. We leveraged the nanopore Cas9 targeted sequencing[47] method to target active retrotransposons in the human genome in a discovery-based approach. To our knowledge, this is the first

application of this method for repetitive mobile element detection. Our experiments indicate that by utilizing guides targeted to specific subfamily sequences, both reference and non-reference insertions can be efficiently enriched and mapped with multiple supporting reads on even the smallest of nanopore sequencing flow cells. Moreover, we demonstrate that individual sequencing experiments readily capture a majority of reference and non-reference elements. In both pooled and single element experiments, MEIs of five subfamilies are robustly enriched, suggesting that this method is widely applicable across mobile elements, and most suitable for high copy genomic elements. In addition, nanopore sequencing offers the ability to detect DNA modifications like 5mC. Even though we obtained nanopore reads with two directions, of the reads belonging to captured full-length L1Hs, 65% extend across the consensus L1 sequence and beyond the L1Hs promoter regions in the $3'->5'$ direction, indicating the ability to investigate L1Hs promoter methylation in Cas9 targeted nanopore sequencing experiments. We further examine CpG methylation profiles of full-length reference and non-reference MEIs (Supplementary Fig. 11), showing consistent results with a prior study[45]. Guide RNA design process is straightforward, and targeting elements based on subfamily nucleotide differences captures both reference and non-reference elements, with negligible loss of sequencing to related close subfamilies or off-targeting. Unlike other Cas9 targeted enrichment experiments, on-target (reference and non-reference) rates for our method are comparatively higher, exceeding 50% in some cases. While this is likely a consequence of the number of genomic copies of the targeted element and, to a lesser extent, the fidelity of the guide sequence, it reiterates this method is particularly suitable for MEI discovery.

The preponderance of uncharacterized MEIs, taken together with their potential contribution to genomic variation and disease, emphasizes the critical need for efficient mobile element detection strategies. Our experiments using Cas9 enrichment and nanopore sequencing can quickly map active mobile elements in the human genome, as well as their larger genomic context. As we expected, in the nanopore-specific MEIs captured by our experiments, 15 out of 17 have reference *Alu*, L1, or LTR regions flanked by the insertion site. The assembly method or PacBio subread mapping (usually shorter than nanopore reads) could find difficulties in these reference repeat regions, where the Cas9 target method with nanopore sequencing could overcome the obstacles (with read length N50 ranged from 14.9 to 32.3 Kbp in the 17 flow cells of our experiments, Supplementary Data 3). Although this is only a handful of overlooked insertions, it is a surprising result from such an extensively sequenced genome as GM12878. As we have shown previously, upwards of 50% of MEIs are missed in data generated from short read sequencing approaches[29]. Taken together with our results showing near complete saturation for MEIs in a single enrichment experiment, we anticipate that this may be highly effective at mapping patient samples, where a substantially larger proportion of MEI sites may differ from the GM12878 reference. In addition, larger nanopore sequencing platforms and multiplexed patient samples for pooled enrichments would streamline processing and maximize cost efficiency.

Since Cas9 enrichment has been used to target rare rearrangement events[71], implementation of this approach to detect unique, de novo, or somatic mobile element insertions is a feasible endeavor. This is emphasized by mounting evidence of MEIs escaping repression in embryogenesis, such as 5′ truncated LINE-1s dodging repression by YY1 in neurons[72]. Nanopore-based identification of particularly rare insertion events, such as mosaic MEIs, is plausible. Recent work using whole genome sequencing of neuronal cells has demonstrated lineage tracing of retrotransposition events in early embryogenesis[73]. The application of targeted enrichment approaches to genetic mosaicism may improve the rate and depth at which this type of variation is sequenced. However, efficiently detecting genetic mosaicism through Cas9 enrichment may require larger sequencing platforms or methodological innovations, as the rarity of target sites in a sample increases.

While we surmise that the vast majority of MEIs can be captured using this approach, it is important to recognize that some genomic locations may persistently conceal recently transposed elements. Centromeric regions and long palindromic repeats are examples of complex genomic features that could be recalcitrant to MEI discovery[74]. With N50s of more than 25 kb, we observed some MEI signals in centromeric and highly repetitive, palindromic regions from our nanopore sequencing reads. However, these regions still complicate mapping, requiring substantially longer sequencing and comprehensive analysis to confidently pinpoint elusive insertions[75]. Merging the enrichment experiments discussed here with improved commercial kits and extremely high molecular weight genomic DNA, may be critical for preserving the extremely long fragments necessary to map MEIs in complex genomic landscapes. The importance of mobile element activity in shaping the genomes they inhabit cannot be overstated. Even beyond the scope of the human genome, mobile element activity plays an intricate role in evolution across many organisms[76,77]. Accelerated discovery of active mobile elements and other repetitive genetic elements will expand our understanding of their contributions to phenotypic diversity in genomes from every form of life.

## Methods

**Cell culture, counting, and genomic DNA isolation**. The following cell lines/DNA samples were obtained from the NIGMS Human Genetic Cell Repository at the Coriell Institute for Medical Research: GM12878, GM12891, GM12892. Each cell line was cultured at 37 °C, 5% $CO_2$ in RPMI 1640 media (ThermoFisher, 11875093) with L-glutamine, and supplemented with 15% fetal bovine serum (ThermoFisher, 10437028) and 1× antimycotic–antibiotic (ThermoFisher, 15240112). Cells were regularly passed and the media replenished every 3 days.

High molecular weight genomic DNA was isolated from GM12878, GM12891, and GM12892 using a "salting out" method[78] with the following modifications. Lymphoblastoid cells were harvested from culture and centrifuged at 500×g for 10 min at 4 °C. Each pellet was washed in 4 °C 1× phosphate buffered saline (PBS) and cell number was counted using the Countess II. Washed cell pellets were resuspended in 3 mL of nuclei lysis buffer (10 mM Tris-HCl pH 8.2, 400 mM NaCl, and 2 mM EDTA pH 8.2). Two hundred microliter of 10% SDS was added to the resuspension and inverted until evenly mixed. Fifty microliter of RNase A (10 mg/mL) was added and the lysate was rotated at 37 °C for 30 min followed by addition of 50 μL of proteinase K (10 mg/mL) and rotation at 37 °C overnight. One milliliter of saturated NaCl solution was added to the lysate and mixed by handheld shaking until evenly mixed. The sample was then centrifuged at 4000×g for 15 min at room temperature. The supernatant was transferred into a new 1.5 mL microcentrifuge tube. Two volumes of 100% ethanol were added to the supernatant and the tube was inverted approximately 20 times, or until the precipitate coalesced. The precipitate was isolated via spooling with a sterile p10 pipette tip and resuspend in a sufficient volume of 1× TE buffer (~250–500 μL, depending on starting amount of cell material). Genomic DNA was passed through a 27G needle three times and stored at 4 °C. The DNA concentration was measured using a Qubit 3 Fluorimeter and the dsDNA Broad Range Assay kit (ThermoFisher, Q32850)

**Design of unique guide RNAs for L1Hs, *AluYa*, *AluYb*, SVA_F, and SVA_E**. To maximize the enrichment performance for each MEI subfamily, the guide RNA (gRNA) candidates were designed to bind to the unique sequences within each subfamily. A pairwise comparison was conducted for the target MEIs with other, non target subfamilies. The consensus sequences for each target MEI subfamily were obtained from Repbase[79], namely L1Hs in the L1 family, *AluYa5* and *AluYb8* in the *Alu* element family representing the *AluYa* and *AluYb* subfamily, and SVA_E and SVA_F in the SVA family. These subfamilies account for over 80% of currently active mobile elements in the human genome[48–50]. The consensus sequences of L1PA2, *AluY*, primate *Alu*, and SVA_D were retrieved from Repbase and included as outgroups in the comparison analysis. Furthermore, *AluYa5* was added as an outgroup in the design of the gRNA for *AluYb*, *AluYb8* for *AluYa*, SVA_E for SVA_F, and SVA_F for SVA_E to avoid enrichment across target MEIs.

Guide RNA target sites (20 bp sgRNA + 3 bp NGG PAM site) for *S. pyrogenes* Cas9 were identified that are within unique MEI regions to obtain optimal guide candidates. Jellyfish2.0[80] was utilized to create a *k*-mer ($k = 23$) index for the sequences of these unique regions, and 23mers with a 5′ "CC" or 3′ "GG" were

selected as gRNA candidates. The frequency of gRNA candidates and the three base substitution options in the "NGG" PAM site for each candidate in the reference genome was calculated to confirm that, the number of unique guide sequences is similar to the genomic reference MEI sequence frequency (Supplementary Data 1 and Supplementary Figs. 1 and 2). The guide RNA candidates of AluY and SVA with unique sequences into were categorized into different tiers: Tier0, sequence has subfamily-specific bases in GG/CC of the PAM site; Tier1, the frequency of 23mer falls into a reasonable range (<2-fold of target MEI frequency) in reference genome; Tier2, the sequence has only subfamily-specific bases at the N site of the PAM or the frequency of the 23mer falls out of a reasonable range. A gRNA sequence falling in Tier0 was considered an ideal candidate.

**On-target boundary calculations for MEIs.** Using the final gRNA selection as a reference, an upper-bound, lower-bound, and intermediate value of the theoretical numbers of target MEIs could be estimated. The lower-bound for target MEIs was defined as a MEI sequence that contains the sequence the gRNA binds to with 100% (or 23 bp) matched sequence, the intermediate bound allows for ≤3 bp mismatch or gap between the gRNA sequence and the matched MEI sequence, and the upper bound is a gRNA that aligns with more than 60% matched sequence (or ≥14 bp) to the MEI.

**In vitro transcription of guide RNA and Cas9 ribonucleoprotein formation.** Single stranded DNA oligos were designed using the EnGen sgRNA Designer tool (https://sgrna.neb.com/#!/sgrna, New England Biolabs) and purchased from IDT (Integrated DNA Technologies) to be used in the EnGen sgRNA Synthesis Kit (New England Biolabs, E3322S). Lyophilized oligos were resuspended in molecular biology grade water to a concentration of 100 μM, and 1:10 dilutions were made for working stocks. Each reaction was set up containing 10 μL of EnGen 2× sgRNA Reaction Mix (S. pyrogenes), 0.5 μL of 100 mM DTT, 2.5 μL of 10 μM oligonucleotide, 2 μL of EnGen sgRNA Enzyme Mix, and brought to 20 μL total with PCR grade water. The reactions were incubated at 37 °C for 30 min to 1 h. To degrade leftover DNA oligonucleotides, the reaction volume was adjusted to 50 μL using PCR grade water, and 2 μL of DNase (New England Biolabs, E3322S) was added to the sample and incubated for 15 min at 37 °C. The sgRNA was purified by adding 200 μL of Trizol and 50 μL of chloroform to the sample, vortexed to mix and centrifuged at 20,000×g at room temperature. The aqueous layer was removed and placed into a new 1.5 mL microcentrifuge tube and extracted again using 50 μL of chloroform. The aqueous layer was removed and placed into a new 1.5 mL microcentrifuge tube and ethanol precipitated in two volumes of 100% ethanol, and sodium acetate was added to a final concentration of 0.3 M. The sample was centrifuged at max speed at 4 °C for 30 min. The RNA pellet was washed with 70% ethanol, air dried, and resuspended in 10 μL of PCR grade water. RNA concentration was measured using the Qubit RNA BR Assay Kit (ThermoFisher, Q10211). Fresh guide RNA was transcribed for every experiment, and prepared no more than a day in advance.

The Cas9 ribonucleoprotein (RNP) was formed by combining 850 ng of in vitro transcribed guide RNA, 1 μL of a 1:5 dilution of Alt-R S.p.Cas9 Nuclease V3 (Integrated DNA Technologies, 1081058), and 1× Cutsmart buffer (New England Biolabs, B7204S) in a total of 30 μL. To allow for sufficient RNP formation, the reaction was incubated at room temperature for 20 min.

**Cas9 enrichment for L1Hs on a MinION flow cell.** To perform a Cas9 sequencing enrichment for L1Hs, a modified Cas9 enrichment experiment was performed[47]. Three identical aliquots of 10 μg of GM12878 genomic DNA were exhaustively dephosphorylated in a total volume of 40 μL, with 1× Cutsmart buffer, 6 μL of Quick CIP (New England Biolabs, M0525S), 10 μg of gDNA, and H2O for 30 min at 37 °C. The Quick CIP was heat inactivated at 80 °C for 20 min. Twenty microliter of RNP (Cas9 + gRNA) was added to the reaction along with 2 μL of Taq polymerase (New England Biolabs, M0273L) and 1.5 μL of 10 mM dATP. The reaction was mixed by tapping and incubated at 37 °C for 30 min for Cas9 cleavage, and 72 °C for 10 min for monoadenylation. Following monoadenylation, each reaction was combined with 50 μL of ligation mix: 25 μL Ligation Buffer (LNB; Oxford Nanopore Technologies, SQK-LSK109), 5 μL of Adapter Mix X (AMX; Oxford Nanopore Technologies, SQK-LSK109), 12.5 μL of T4 DNA ligase (New England Biolabs, M0202M), and 5 μL of nuclease-free water. The nanopore adapters were ligated to the genomic DNA at room temperature for 30 min on a tube rotator. Once completed, the ligations were diluted with 1 volume of 1× TE buffer (100 μL). Sixty microliter of SPRI beads (Beckman Coulter, B23317) were added to the adapter ligated samples and incubated at room temperature for 10 min with rotation, and for another 5 min without rotation. Beads were immobilized using a magnet and the supernatant was removed. Immobilized beads were resuspended with 200 μL of room temperature L fragment buffer (LFB; Oxford Nanopore Technologies, SQK-LSK109). At this step, the resuspended beads from the three samples were pooled into one Eppendorf tube. The magnet was applied again to immobilize the beads and remove the supernatant and the wash was repeated. Washed samples were pulse spun on a tabletop centrifuge for 1 s to collect beads at the bottom. Residual LFB was aspirated with a pipette. Beads were resuspended in 16.8 μL of Elution Buffer (EB; Oxford Nanopore Technologies, SQK-LSK109) and incubated at room temperature for 10 min. Following the

elution, the magnet was applied and the supernatant was collected and placed into a sterile Eppendorf tube. In some sample preparations, the adapter ligated library eluted in the last step may be viscous and the beads will resist immobilization on the magnet. A maximum speed centrifugation step prior to applying the magnet will help to immobilize the beads. Once the supernatant was separated from the beads into a sterile Eppendorf tube, 26 μL of Sequencing Buffer (SQB; Oxford Nanopore Technologies, SQK-LSK109) was added and placed on ice until the sequencing flow cell was prepared. Immediately prior to loading of the sample, 0.5 μL of Sequencing Tether (SQT, Oxford Nanopore Technologies, SQK-LSK109) was added along with 9.5 μL of Loading Beads (LB; Oxford Nanopore Technologies, SQK-LSK109). The sample was mixed evenly by pipetting with a p20 and loaded onto the sequencing platform.

**Pooled Cas9 enrichment for L1Hs, AluYb, AluYa, SVA_F, and SVA_E in GM12878 (MinION).** Five parallel Cas9 enrichment experiments were performed for the five MEI subfamilies in GM12878 for a pooled sequencing run. Five separate aliquots of 10 μg of genomic DNA were dephosphorylated in 40 μL total (30 μL of gDNA, 4 μL of 10× CutSmart, 6 μL of Quick CIP) for 25 min at 37 °C then heat inactivated at 80 °C for 5 min. Following the heat inactivation, each dephosphorlyated genomic DNA sample was combined with 20 μL of Cas9 RNP, 1 μL of Taq polymerase, and 1 μL of 10 mM dATP. After briefly mixing by tapping, the reaction was incubated at 37 °C for 30 min to enable Cas9 cleavage, then incubated to 75 °C for monoadenylation by Taq polymerase. The Cas9 digested and monoadenylated samples were pooled into the ligation reaction (164 μL of Custom LNB, 10 μL of AMX, 20 μL of T4 DNA ligase, and 164 μL of nuclease-free water) and rotated at room temperature for 30 min. One volume of 1× TE buffer was added to the ligation and mixed by inversion approximately ten times, or until evenly mixed. 0.3× sample volume of SPRI beads (394.8 μL) was added and incubated at room temperature with rotation for 5 min. The beads were immobilized using a magnet and washed twice with 100 μL of room temperature LFB. After the final wash, the beads were pulse spun for 1 s in a table top centrifuge, immobilized on a magnet, and residual LFB was removed. The washed beads were eluted in 13 μL of EB for 10 min at room temperature and removed using a magnet. The supernatant was collected and combined with 26 μL of SQB. The library was incubated on ice until the flow cell was prepared. 0.5 μL of SQT and 9.5 μL of LB were added to the library before the loading onto the flow cell.

**Cas9 enrichment for single MEI subfamily on a Flongle flow cell.** Ten microgram of purified genomic DNA was dephosphorylated using 4 μL of Quick CIP in 1× Cutsmart buffer and brought to a total reaction volume of 40 μL, then incubated for 30 min at 37 °C. The sample was then incubated at 80 °C for 5 min to inactivate the Quick CIP. One microliter of Taq polymerase, 1 μL of 10 mM dATP, and 20 μL of the corresponding Cas9 RNP (targeting L1Hs, AluYb, AluYb, SVA_F, or SVA_E), was added to the dephosphorlyated genomic DNA, gently mixed, and incubated at 37 °C for 30 min, followed by a 10 min incubation at 75 °C. The sample was added to the ligation solution (25 μL of custom LNB, 6 μL of T4 DNA ligase, 5 μL of AMX, and nuclease-free water to 100 μL total), and incubated at room temperature for 20 min with rotation. The ligation was mixed with 1 volume (100 μL) of 1× TE buffer and mixed by inversion approximately ten times, or until evenly mixed. SPRI beads were added to a final 0.3× (60 μL) to the sample volume (200 μL) and the sample was rotated at room temperature for 5 min. The SPRI beads were immobilized on a magnet and washed twice with 100 μL of room temperature LFB. After the final wash, the beads were pulse spun for 1 s on tabletop centrifuge and residual LFB was removed. The beads were resuspended in 9 μL of EB and incubated at room temperature for 10 min. After the elution, the beads were immobilized on a magnet and the supernatant was transferred to a new 1.5 mL microcentrifuge tube. Thirteen microliter of SQB was added to the supernatant and this library was placed on ice until the flow cell was prepared. Before loading the sample onto the flow cell, 0.5 μL of SQT and 9.5 μL of LB were added and mixed by gentle tapping.

**Cas9 enrichment for L1Hs in trio (MinION).** To detect L1Hs in the lymphoblastoid trio cells (GM12878/91/92), a modified Cas9 enrichment assay, originally described by Gilpatrick et al.[47], was performed. Ten microgram of genomic DNA for each genome (30 μg total) was exhaustively dephosphorylated using 6 μL Quick CIP for 45 minutes at 37 °C, and heat inactivated at 80 °C for 20 min. Twenty microliter of the RNP (Cas9 and sgRNA) was added to the dephosphorylated genomic DNA along with 2 μL of Taq DNA polymerase and 1.5 μL of 10 mM dATP. The reaction was incubated at 37 °C for 30 min, then 72 °C for 10 min. Each genomic DNA reaction was combined with an equal volume (50 μL) of ligation mix for nanopore adapter ligation: 25 μL Ligation Buffer (Oxford Nanopore Technologies, EXP-NBD104), 5 μL of AMII (Adapter Mix II; Oxford Nanopore Technologies, EXP-NBD104), 12.5 μL of T4 DNA ligase, 2.5 μL of the barcode (NB01/02/03; Oxford Nanopore Technologies, EXP-NBD104), and 2.5 μL of nuclease-free water, and incubated at room temperature for 30 min on a tube rotator. After adapter ligation, an equal volume of 1× TE buffer was added to the reaction, and SPRI beads were added to a final 0.3× (~60 μL). The library was incubated at RT for 10 min with rotation, and 10 min without rotation for a total of 20 min to allow for DNA binding to the SPRI beads. The beads were washed twice with 200 μL of L-Fragment Buffer (LFB; Oxford Nanopore

Technologies, EXP-NBD104). The uniquely barcoded samples were pooled by combining the resuspended beads in the first wash, then washed again. The washed beads were resuspended in 16.8 µL of the Elution Buffer (EB; Oxford Nanopore Technologies, EXP-NBD104). Resuspended beads were incubated at room temperature for 10 min. Following the incubation, the beads were collected with a magnet and the supernatant collected into a separate 1.5 mL microcentrifuge tube and placed on ice. The sample was prepared for sequencing by adding 26 µL of Sequencing Buffer (SQB; Oxford Nanopore Technologies, EXP-NBD104) and kept on ice while the flow cell was primed. 0.5 µL of Sequencing Tether (SQT; Oxford Nanopore Technologies, EXP-NBD104) and 9.5 µL of Loading Beads (LB; Oxford Nanopore Technologies, EXP-NBD104) were added after flow cell priming, before the sample was loaded onto the flow cell.

**Nanopore flow cell preparation, sequencing, base-calling, and cleavage-site analysis.** A MinION flow cell was purchased from Oxford Nanopore Technologies and stored at 4 °C per manufacturer's instructions. The Ligation Sequencing Kit (SQK-LSK109) and Native Barcoding Kit (EXP-NBD104) were used to prepare the pooled libraries. Upon arrival and prior to usage, MinION flow cell QC was performed using the MinKNOW software. No appreciable loss of active pores was noted during the run. Prior to loading the library, the MinION was flushed with 800 µL of FLB in the priming port, followed by priming of the flow cell with 200 µL of 0.5× SQB diluted with water.

Flongle flow cells were purchased from Oxford Nanopore Technologies in batches and stored at 4 °C. Flow cells were QC'd upon arrival and the number of active pores were noted. Flongles to be used in sequencing were QC'd immediately before use to assess pore loss during the storage period.

Base-calling was processed by Guppy 4.0.15 (Oxford Nanopore Technologies) using the high accuracy, modified base model (dna_r9.4.1_450bps_modbases_ dam-dcm-cpg_hac.cfg). Porechop[81] was used to trim nanopore adapters and barcodes from the reads with Q-score > 7, as well as demultiplex the reads in the pooled sample MinION run.

To determine cut site preferences, reads were aligned to the consensus sequence of each mobile element class that were investigated; L1Hs, AluYa5, AluYb8, SVA_E, and SVA_F. Only the first 80 base pairs of each read were used for alignment to focus on the cut site region, and to ensure that the beginning of the reads aligned correctly. Pairwise alignments were performed using the Biopython Bio.Align package[82] with FASTA files as input. Each read was aligned to the mobile element consensus sequence as well as the reverse complement to determine sequence orientation. To obtain high-confidence alignments, strict gap penalties were enforced (open gap: −10, extend gap: −5). In order for an alignment to be considered for cut-site analysis, it had to meet two criteria; the alignment had to start at the very first base of the read, and needed to have an alignment score of at least 100. The 5′ ends of reads meeting this criteria were then used to estimate cleavage site location.

**Nano-Pal for detection and refinement of MEIs from nanopore Cas9 enrichment.** To resolve both the reference and non-reference MEI signals from nanopore Cas9 enrichment, we developed a computational pipeline, Nano-Pal, to analyze the nanopore reads and customized it for different MEI subfamilies (Fig. 1c, d). Information of the potential targeting MEI signals was obtained by Nano-Pal scanning through both sides (100 bp bin size) of all quality-passed nanopore raw reads using BLASTn[83,84]. Next, it aligned the reads to the reference genome (GRCh38) using minimap2[85] and discarded reads with low mapping quality (MAPQ < 10). The aligned reads were screened by RepeatMasker[67] and the pre-masking module in PALMER[29] to bin them into different categories: reads with reference MEI signals, reads with non-reference MEI signals, and off-target reads. All reads that were reported by the PALMER pre-masking module fell into the on-target non-reference MEI category. If reads were not reported by PALMER, but were annotated by RepeatMasker, they fell into the reference MEI category. They will be further classified as on-target, close-target, and off-target reads depending on where they mapped to the reference regions. For L1Hs experiments, the reads mapped to the reference L1PA are considered as close-target and the ones mapped to other reference L1 are considered as off-target. For AluY experiments, the close-target reads are those that mapped to other reference AluY besides AluYb and AluYa and the off-target reads are defined when mapped to other reference Alu elements besides the reference AluY. For SVA experiments, the close-target reads are the ones mapped to other reference SVAs besides SVA_F and SVA_E and no off-target reads were defined when they have the MEI signals. Any remaining reads with no MEI signals were classified as off-target reads as well. The reads in the first and second category were then clustered into non-reference MEIs and reference MEIs, respectively. Nano-Pal was performed for each Flongle and MinION flow cell separately for GM12878, GM12891, and GM12892.

**GM12878 trio data, reference genome, and reference MEI information.** We obtained Pacific Bioscience (PacBio) long-read CLR sequencing data from Audano et al.[60] for the GM12878 genome (50× coverage). The 30× Illumina NovaSeq sequencing data for GM12878 and the related samples (GM12891 and GM12892) were obtained from the 1000 Genomes project phase 3 sample set, which were generated at the New York Genome Center (ftp://ftp.1000genomes.ebi.ac.uk/vol1/

ftp/data_collections/1000G_2504_high_coverage/)[57,86]. All analyses in this project were carried out using the GRCh38 (GRCh38+decoy) reference genome obtained from the 1000 Genomes Project (ftp://ftp.1000genomes.ebi.ac.uk/vol1/ftp/ technical/reference/GRCh38_reference_genome/). Information of reference MEIs, including the five target subfamilies, were obtained from RepeatMasker[67].

**Enhanced PALMER for resolving non-reference MEIs from whole-genome long-read sequencing.** We developed an enhanced version of PALMER (Pre-mAsking Long reads for Mobile Element inseRtion)[29] in this study to detect non-reference MEIs across the long-read sequenced genomes (https://github.com/mills-lab/PALMER). Reference-aligned BAM files from long-read technology were used as input. Known reference repetitive sequences (L1s, Alus, or SVAs) were used to pre-mask portions of individual reads that aligned to these repeats and also utilized in the Nano-Pal pipeline. After the pre-masking process, PALMER searched sub-reads against a library of consensus mobile element sequences within the remaining unmasked sequences and identified reads with a putative insertion signal (including 5′ inverted L1 sequences, if available) as supporting read candidates. PALMER opens bins 5′ upstream and 3′ downstream of the putative insertion sequence for each read and identifies hallmarks of mobile elements, such as target site duplication (TSD) motifs, transductions, and poly(A) tract sequences. All supporting reads are clustered at each locus and those with a minimum number of supporting events are reported as putative insertions.

To improve the accuracy of non-reference MEI sequences derived from individual subreads, which tend to have lower per-read base-pair accuracy, local sequence alignments, and error correction strategies were performed. Error correction was conducted by CANU[87] (ver2.2) using default parameters on the subreads with MEI signals reported by PALMER, allowing the generation of error-corrected reads that served as inputs for local realignment using minimap2. A second-pass of the PALMER pipeline then was executed using these locally aligned error-corrected reads to generate a high-confidence call set of germline non-reference MEIs. CAP3[88] was used with default parameters to assemble all MEI sequences reported by the second-pass of the PALMER pipeline to generate a high-confidence consensus contig for each non-reference MEI event.

**MEI callsets in orthogonal short-read and long-read data.** As there is no public long-read data available for GM12891 and GM12892, we used the Mobile Element Locator Tool (MELT) to identify non-reference MEIs in the short-read Illumina sequencing data for the GM12878 trio[27,68] as a benchmark set in the trio analysis. We applied the enhanced version of PALMER and carried out the non-reference MEI calling in GM12878. To generate a more comprehensive callset of non-reference MEIs in GM12878, the Phased Assembly Variant (PAV) caller was included (https://github.com/EichlerLab/pav), which can discover genetic variants based on a direct comparison between two sequence-assembled haplotypes and the human reference genome[46]. The callset by PAV for GM12878 was generated from the PacBio HIFI sequencing data after haplotype-assembly. A "PacBio-MEI" callset in GM12878 was generated by applying the union set of the mapping-based PALMER callset and the assembly-based PAV callset, both of which resolved the MEIs from PacBio long-read sequencing data. The details of MEI merging and subfamily defining strategy for the two approaches are described in a prior study[46].

**Inspection and validation of nanopore-specific non-reference MEIs.** All non-reference MEI calls were further intersected with the PacBio-MEI set and classified as known non-reference calls and potential nanopore-specific non-reference calls. A filtering module, an empirical curation of read-depth (<2-fold difference) within the 500 bp bin of the insertion site from public data[60], and manual inspection were applied to exclude false-positive (FP) signals in the nanopore-specific non-reference MEIs. All potential nanopore-specific non-reference MEIs were classified into three categories: (a) true positive (TP) non-reference event, (b) FP non-reference event, and c) an ambiguous event (Supplementary Data 8). The category was further defined as TP missed by the PacBio sequencing data, TP missed by the PacBio mapped-based and assembly-based pipelines but with PacBio read signals, or TP redundant with the called non-reference one. Category (b) was broken down into three subcategories; FP redundant with the called reference event, FP targeting on the other off-target reference repeat, or ambiguous. All subcategories in (a), and the first (b) subcategory, were on-target reads with or without correct annotations. For example, one FP nanopore-specific non-reference call originated from reads targeted to reference MEIs, yet was categorized as non-reference due to a mapping error introduced by flanking structural variations (deletions, duplications, or inversions) (Supplementary Data 8). A further in-depth inspection was employed for the non-reference calls categories as nanopore-specific. Each call has been inspected by two sections: (a) general information including TRPT hallmarks (TSD motifs, poly(A), EN Cleavage site, and empty site sequence), length, strand, genotype, and population frequency in 32 genomes reported by Ebert et al.[46], and (b) IGV screenshot in a range of genomic region with genomic content annotation. The TRPT hallmarks were identified by Cas9 target nanopore reads or from PacBio assembled contigs (Supplementary Data 9). A recurrence plot analysis was employed for further in-depth validation as well. For the recurrence plot analysis, a region of one sequence (X-axis) is compared to another sequence (Y-axis) and small (i.e., 10 bp) segments that are identical between the two sequences are

denoted with a plotted point. Thus, a continuous diagonal line comprising multiple points indicates portions of the compared sequences that are identical. By comparison, gaps, and shifts from the diagonal denote an insertion or deletion in one sequence relative to the other.

**Non-reference MEIs captured by nanopore Cas9 enrichment sequencing in GM12878**. The flow cell runs for each MEI subfamily were merged to investigate the read coverage enrichment performance and generate the final non-reference MEIs in GM12878. For the saturation analysis, the flow cells were ranked by the number of on-target reads. Reads from the flow cells were then added and merged, one flow cell at a time, based on the above ranking. Non-reference calls for L1Hs, *Alu*Yb, *Alu*Ya, SVA_F, and SVA_E were resolved by Nano-Pal and our validation process after every merging instance. By merging all batches for each subfamily, a final non-reference MEI callset in GM12878 captured by nanopore Cas9 enrichment approach was produced.

**Analysis of L1Hs CpG methylation**. Nanopolish[89] was used to call methylation on pooled GM12878 MinION and Flongle runs. Reference L1Hs methylation profiles were generated using methylartist (https://github.com/adamewing/methylartist), where methylation was aggregated across L1HS intervals from RepeatMasker[67]. To generate profiles for non-reference elements, we built contigs using reads supporting full-length L1Hs and ±100 kb of flanking sequences. Reads within 500 bp of insertion sites were then extracted from the merged data and aligned to the constructed contigs using minimap2[85]. These alignments and reads were then used for methylation calling with nanopolish. The data from nanopolish was aggregated across L1Hs sequence in the constructed contigs and used for locus and consensus plotting using methylartist.

**Reporting summary**. Further information on research design is available in the Nature Research Reporting Summary linked to this article.

## Data availability

The nanopore sequencing data for the Cas9 targeted enrichment of MEIs in this study are available in the SRA repository under BioProject accession PRJNA699027.

## Code availability

All scripts and pipelines in this publication, including Nano-Pal, are available on GitHub: https://github.com/Boyle-Lab/NanoPal-and-Cas9-targeted-enrichement-pipelines[90]. The enhanced version of PALMER is available at https://github.com/mills-lab/PALMER[91].

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

## Acknowledgements

We thank Dr. Scott Devine, Dr. Qihui Zhu, and Dr. Charles Lee for providing the MELT callset for GM12892 and GM12891. We thank Jixin Guan for their help with improving the performance of the PALMER software. This research was supported by the National Institutes for Health (NIH) under award no. R21HG011493 to A.P.B. and R.E.M. T.M. was supported by T32GM007544. C.C. was supported by the University of Michigan Rackham Merit Fellowship and the Training Program in Bioinformatics (T32GM070449).

## Author contributions

A.P.B., R.E.M., W.Z., and T.M. conceived the project. J.S. established and cultured cell lines and isolated gDNA. T.M. performed the Cas9 targeted enrichment and nanopore sequencing. W.Z. developed the NanoPal pipeline and PALMER software. W.Z., C.C., and C.M. performed computational analysis. All authors guided the data analysis strategy. A.P.B., R.E.M., W.Z., and T.M. wrote the manuscript. All authors edited the manuscript. All authors read and approved the final manuscript.

## Competing interests

The authors declare no competing interests.
