## [Peer Review File · Nature Communications]

Reviewers' Comments:

Reviewer #1:

Remarks to the Author:

McDonald and colleagues use Cas9 enrichment and updated computational strategies to capture young mobile element insertions (MEIs) in the GM12878 cell line. While this approach shows promise, it seems to also perform quite variably in terms of enrichment and on/off target ratio between runs. Cas9 enrichment (Gilpatrick et al, Nat Methods, 2020) and Oxford Nanopore (ONT) long-read sequencing to detect MEIs (Ewing et al. Mol Cell, 2020) have both been reported recently. Therefore one would fairly expect data to show the present method works well enough in terms of cost:benefit to be broadly adopted. Despite its merits and the general interest in this technology, the present approach falls short of this.

Major criticisms:

1) The authors test multiple conditions for MEI enrichment and sequencing, which was good to see. However, the headline performance statistics made in the Abstract, literally noted as "remarkable", do not seem to be representative of what was consistently achieved. What were the reproducible on/off target ratios for each MEI family? This is a crucial part of deciding whether enrichment is worthwhile from a cost:benefit tradeoff standpoint, even more so than coverage or throughput. Parsing the various statistics in the results section, each subject to different parameters, it was hard to know how robust and reproducible enrichment was. For a reader to decide whether to test this method, they should know realistically how well it is expected to perform - the Abstract statistics seen to be best case.

2) Twenty-one previously overlooked non-reference MEIs in GM12878 are mentioned in the Abstract and elsewhere. How many of these MEIs carry TPRT hallmarks? What are their TSD sequences, polyA tracts, EN motif? Are they sequenced in their entirety (i.e. by a spanning read)? The authors (Zhou et al NAR 2020) previously used PacBio long read sequencing to great effect to find this type of "occult" MEI - it is surprising but not unrealistic that ONT sequencing could find more of these than PacBio sequencing applied to the same sample. However, conclusive evidence is required as to whether these MEIs are true or false positives. Relying so heavily on junction-spanning, but not necessarily MEI-spanning reads, seems a potential source of false positives.

3) I felt the authors could perhaps have discussed more plainly the limitations of this study and method. MEIs have been mapped with ONT sequencing applied to human tissues elsewhere (Ewing et al, above) and this really needs to be accounted for correctly in several places in the text. That the MEI mapping done by Ewing et al is not mentioned at all is difficult to understand, at least from a scientific standpoint. Semantic claims of novelty in the Discussion are unnecessary as well. Also, as the authors bring up potential for mapping MEIs in neurodevelopment, they should cite the two studies reporting these with clear evidence (Evrony et al Neuron 2015; Sanchez-Luque et al Mol Cell 2019).

4) More importantly, some discussion of technical limitations alongside merits would be helpful to readers. The necessary focus on L1HS guides at the 3' end (though there is some subfamily diversity towards the 5' end) leaves open two technical limitations: firstly, how often are reads spanning full-length L1HS MEIs recovered and, secondly, can L1HS promoter methylation be surveyed with this approach? Being able to survey methylation and other DNA modifications is a major advantage of ONT sequencing, and particularly Cas9 enrichment. This would have been useful to highlight even if an analysis was not performed.

Reviewer #2:

Remarks to the Author:

Review of McDonald et al, "Cas9 targeted enrichment of mobile elements using nanopore sequencing"

Synopsis:

This well-written manuscript describes the application of Cas9-based long-read targeted sequencing of mobile element insertions. The work is unique in that the method employs hybridization-free and PCR-free sequencing of transposable elements. Characterization of such elements using long-reads provides many advantages over short-read based assembly approaches. For example, the distance between MEI elements using this approach can approach 100kb lengths due to the ultra-long read lengths of nanopore sequencers. Cleavage of the MEI only has to occur once and nanopore sequencing can initiate at that position, which is a key advantage of the technique.

In addition to the experimental work, the authors have developed a computational pipeline (Nano-Pal) to characterize known and novel elements, particularly elements that are embedded in complex genome regions. The research problem is interesting, and the authors chose a good model system with plenty of prior characterization (GM line, including trio) for method development.

The manuscript is technically sound overall. However, stronger validation of the novel MEIs using an orthogonal measurement (e.g., PCR-based targeting, with primers that flank the non-reference MEI) would strengthen the paper. The error-prone nature of single-pass nanopore sequencing brings up the question of the resolvability of closely related transposable element family members, given the challenge of deriving consensus sequences from ONT reads.

The authors present sufficient detail to reproduce the work. The statistical analyses and evidence was also adequate.

Major comments:

1. Figure 2a-c – Unclear on which plotting elements correspond to left versus right Y-axis. Why are the gRNAs for the L1Hs more evenly dispersed compared to Alus and SVAs?
2. Figure 2a-c – What is the reasoning for including the spatial distribution of gRNAs, when a single candidate was chosen? Is this to show additional potential targets along the span of the MEI for future work? Authors should indicate the position of the gRNA selected for this study (e.g., a red arrow).
3. p3 – Suggest a statement in introduction that Cas9 targeting approaches can be combined with any long-read RNA-seq platform (ONT, PacBio).
4. Introduction – It would be helpful if the authors could cite more past work that involve targeted sequencing strategies of repetitive and transposable elements and delineate the distinguishing features and advantages of their approach. Such points could also be made in the Discussion.
5. Discussion, page 13 – GM is well characterized, and the utility of the method is demonstrated by the discovery of 21 non-reference MEIs. Can the authors give an estimate for the number of non-reference MEIs likely discovered for patient-specific genomes? Or common on the potential of the technique? How many non-reference MEIs would be typically found in an uncharacterized genome?
6. Discussion, page 13 – Can the authors comment on the properties of the 21 non-reference MEIs that caused them to be missed by previous approaches?
7. Discussion – Based on the mechanism of the Cas9 enrichment, it seems like this technique could also be integrated with PacBio long-read sequencing, with minor modifications to the adapter ligation and development of a computational pipeline to process the data. The authors should mention this possibility in the introduction and discussion.
8. Discussion – The recent ONT and PacBio systems (e.g., PromethION, Sequel II) are also much higher in throughput (e.g., 4 million reads + in a single flow cell). This should be mentioned in the discussion with an estimate of the impact of 10X higher throughput on targeted MEI analysis. This may become important for increasingly rare targets or scaling up the method.
9. Discussion, page 14 – The authors should outline the remaining limitations/challenges of the approach. In particular, discussion about the reasons for low target enrichment for certain families based on guide specificity, frequency, or any other factors, would be informative to the reader. What are the possibilities for scale-up of this approach and potential use in the clinic? Where are

the limitations in terms of resources (e.g., library prep, sequencing costs)? Though error correction of the reads were done (e.g., CANU), ONT reads are still error prone (e.g., <95% accuracy). Are there closely related MEI families for which there may be more difficulty resolving the subfamilies based on high sequence homology?

Minor comments:

1. p2 – define L1 (as acronym for LINE)
2. p2 – SVA acronym needed
3. Figure 1b – I suggest a split in the Time vs Current plot, because as is, it looks like the sequencing is done contiguously for the left/right MEI sequences, which is not the case.
4. Figure 1c – Suggest specifically defining “MEI signal”. “The yellow bar represents MEI consensus sequence or MEI signals in pairwise comparison of Nano-Pal.” is unclear. I interpret it as the dominant portion (rather than the shorter portion) of the MEI post-Cas9 cleavage. Is there a significance to the MEI element being shorter in “MEI signal on one end”?
5. p3-4 – Please define Nano-Pal either at the end of the introduction or beginning of the results. Nano-Pal is described in Figure 1 without previous description that it is the computational pipeline for analysis of Cas9-enriched ONT data. Some aspects of the introduction to Nano-Pal on page 6 can be moved up for clarity.
6. Supp Figure 3 – What is “maize bar”?
7. Supp Figure 4 legend – “off-targeted” to “off-target”
8. p6, paragraph 2 – What is the significance of the lower lengths for reads with MEIs on both ends? Is this expected based on the spatial distribution of L1Hs across the genome, compared to Alu and SVA elements?
9. What is the reason for the low on-target rates for the SVA family MEIs? Perhaps it would be helpful to add another column to compare the frequency in the genome for each MEI as a function of on-target rates.
10. Supp Figure 5 – Define PAV
11. Discussion – Do the authors have an open source tool for design of gRNAs?
12. p15 – “Two volumes of 100% ethanol...”
13. p15 – “~250-500uL”, uL to microliters or mew symbol
14. p20 – “(MAPQ<10)”

REVIEWER COMMENTS

Reviewer #1 (Remarks to the Author):

McDonald and colleagues use Cas9 enrichment and updated computational strategies to capture young mobile element insertions (MEIs) in the GM12878 cell line. While this approach shows promise, it seems to also perform quite variably in terms of enrichment and on/off target ratio between runs. Cas9 enrichment (Gilpatrick et al, Nat Methods, 2020) and Oxford Nanopore (ONT) long-read sequencing to detect MEIs (Ewing et al. Mol Cell, 2020) have both been reported recently. Therefore one would fairly expect data to show the present method works well enough in terms of cost:benefit to be broadly adopted. Despite its merits and the general interest in this technology, the present approach falls short of this.

We thank the reviewer for critically assessing our manuscript, and we provide a point-by-point response to their comments below.

Major criticisms:

1) The authors test multiple conditions for MEI enrichment and sequencing, which was good to see. However, the headline performance statistics made in the Abstract, literally noted as "remarkable", do not seem to be representative of what was consistently achieved. What were the reproducible on/off target ratios for each MEI family? This is a crucial part of deciding whether enrichment is worthwhile from a cost:benefit tradeoff standpoint, even more so than coverage or throughput. Parsing the various statistics in the results section, each subject to different parameters, it was hard to know how robust and reproducible enrichment was. For a reader to decide whether to test this method, they should know realistically how well it is expected to perform - the Abstract statistics seen to be best case.

We thank the reviewer for their comments. We agree that there is variability in our results, primarily due to differences in pore efficiency between individual flow cells, and we report the results of replication experiments including on/off target ratios for each MEI family across multiple flongles and pooled MinION flow cells in Supplemental Table 3. We find relatively stable on-target rates for a specific guide target, and differences in guide target rates can be expected due to different numbers of genomic integration sites, high numbers of similar target sequences, or guide RNA efficiency variation. For example, the low number of SVAs in the genome resulted in rather low on-target reads, but sufficient coverage to identify most of these elements in the genome. Similarly, the high number of *Alus* in the genome increases the set of near-matched guides and so we have a high background of other *Alus* enriched. We also find a clear (and expected) correlation with read depth and identified MEIs (**Supplemental Table 3** and **Figure 4**). We have now added a discussion of these potential confounders in the Discussion section, and have also now omitted the 'remarkable' comment in the abstract.

We have further conducted a comparison to the whole-genome nanopore sequencing approach reported by Ewing et al. Mol Cell, 2020 to provide an estimate of cost:benefit between targeted flongle/minion and promethion whole genome shotgun sequencing (new **Supplementary Fig.5** and below). When taking total sequenced reads into account, our targeted approach exhibits between a 13.4 to 54 fold increase in the average number of reads per MEI. Furthermore, our read length N50 ranged from 14.9Kbp to 32.3Kbp compared to 5.14Kbp to 10.57Kbp reported in Ewing et al, suggesting that our targeted approach also results in a higher number of MEI spanning reads. Overall, these comparisons indicate that, on the basis of per-base

sequenced, MEI target capture exhibits significant enrichment advantages over whole genome approaches. We have now included a more thorough comparison in the Results section.

Supplementary Fig.5: Number of supporting reads for three categories of non-reference MEI from the Cas9 targeted nanopore sequencing and the whole-genome nanopore sequencing by Ewing et al. 2020.

a, Number of supporting reads for non-reference L1Hs, *AluY*, and SVA in the whole-genome nanopore sequencing (from five PromethION flow cells) and the Cas9 targeted nanopore sequencing (L1Hs from 11 MinION/Flongle flow cells, *AluY* from four from 4 MinION/Flongle flow cells, and SVA from 4 MinION/Flongle flow cells). **b**, Number of supporting reads of non-reference MEIs normalized by the total yield base pairs from flow cells in two studies.

Pg 14 Discussion: However, we observe that some of our MEIs targets (SVAs and *Alus*) have reduced enrichment compared to L1Hs. While we find relatively stable on-target rates for a specific guide target, differences in guide target rates can be expected due to different numbers of genomic integration sites, high numbers of similar target sequences, or guide RNA efficiency variation. For example, the low number of SVAs in the genome resulted in rather low on-target reads, but sufficient coverage to identify most of these elements in the genome. Similarly, the high number of *Alus* in the genome increases the set of near-matched guides and so we have a high background of other *Alus* enriched.

Pg. 7 Cas9 targeted enrichment efficiently captures mobile element signals in nanopore reads: We next assessed the efficiency of our target enrichment of MEIs compared to whole-genome sequencing (WGS) approaches. A recent, related study used a whole-genome nanopore sequencing approach (Ewing et al. 2020) to study MEIs and methylation and provides an excellent benchmark to which we may compare our results. When taking total sequenced reads into account, our targeted approach exhibited between a 13.4 to 54 fold increase in the average number of reads per MEI compared to WGS (**Supplementary Fig.5**). Furthermore, our read length N50 ranged from 14.9Kbp to 32.3Kbp compared to 5.14Kbp to 10.57Kbp reported in Ewing et al, suggesting that our targeted approach also results in a higher number of MEI spanning reads. Overall, these comparisons indicate that on the basis of per-base sequenced, MEI target capture exhibits significant enrichment advantages over whole genome approaches.

2) Twenty-one previously overlooked non-reference MEIs in GM12878 are mentioned in the Abstract and elsewhere. How many of these MEIs carry TPRT hallmarks? What are their TSD sequences, polyA tracts, EN motif? Are they sequenced in their entirety (i.e. by a spanning read)? The authors (Zhou et al NAR 2020) previously used PacBio long read sequencing to great effect to find this type of "occult" MEI - it is surprising but not unrealistic that ONT sequencing could find more of these than PacBio

sequencing applied to the same sample. However, conclusive evidence is required as to whether these MEIs are true or false positives. Relying so heavily on junction-spanning, but not necessarily MEI-spanning reads, seems a potential source of false positives.

In this project, we currently are using a ‘PacBio-MEI’ callset as the benchmark to compare the callset captured by Cas9 targeted nanopore sequencing. As we described in the Methods: *A ‘PacBio-MEI’ callset in GM12878 was generated by applying the union set of the mapping-based PALMER callset and the assembly-based PAV callset, both of which resolved the MEIs from PacBio long-read sequencing data. The details of MEI merging and subfamily defining strategy for the two approaches are described in a prior study (Ebert et al. 2021. Science).* To be noted, PALMER uses aligned subreads to resolve non-reference MEIs, while in Zhou et al 2020 NAR, we employed a fairly stringent criteria in order to obtain the most confident set of non-reference L1Hs in NA12878. PAV is an assembly-based caller and would provide a fine set of non-reference MEIs with the fully resolved sequences from assembled contigs, assuming that the assembly process was working well. However, we observed in some regions that the genome can not be fully resolved by the assembly method in the Ebert et al. 2021 Science paper. We also observed a slight discrepancy between the callsets by PALMER and PAV regarding the MEIs described in **Supplementary Fig.6**. Even though we used the union of PALMER and PAV callsets as the benchmark callset ‘PacBio-MEI’, there are still some calls that could be missed by ‘PacBio-MEI’ possibly due to the bias from PacBio technology or whole-genome sequencing. Thus, the Cas9 target nanopore sequencing for MEI capture could potentially identify complementary calls compared to the ‘PacBio-MEI’ callset, which as the reviewer notes is not unrealistic.

Even though we have manually inspected all 379 potential nanopore-specific non-reference MEIs in this project (**Supplementary Table 7**), we rigorously double-checked the 21 MEIs we believed to be true regarding questions of the TPRT hallmarks and other information concerned by the reviewer. We inspected their TSD motifs, poly(A), EN Cleavage site, empty site sequences, and potential population frequency reported by Ebert et al. 2021. Science. We also inspected their general information with IGV screenshot and genomic information annotated. This information and analyses can be found in the new **Supplementary Table 8**. Eventually, we finalized 12 non-reference L1Hs and 5 non-reference *AluY* as true positives after the analysis (**Supplementary Table 7,8**). We have excluded four non-reference L1Hs events due to lack of support either from TSD motifs or poly(A) tail. These four events could be duplicated sequences harboring an existing L1Hs element, though they still could represent *bona fide* L1Hs insertions. We also plotted the histogram of length of TSD and polyA tail for the seventeen nanopore-specific non-reference MEIs as shown here and in new **Supplementary Fig.10**.

Supplementary Fig.10: Seventeen nanopore-specific MEIs have hallmarks of retrotransposition consistent with bona fide insertions.

We require at least 4 nanopore reads to support the specific non-reference calls. There are 14 out of 17 nanopore-specific calls that can be defined by nanopore reads sequenced from both directions at the guide site, which provide the same information as individual MEI-spanning reads. In addition, as the reviewer notes, there are three non-reference nanopore-specific calls whose TPRT hallmarks can not be defined solely by nanopore reads due to four reads all sequenced in one direction. However, the sequences of these calls can still be recovered by the assembled contigs reported by Ebert et al. 2021. Science, though they were missed by PAV in the NA12878 sample.

As we described in Figure 4, Cas9 enrichment and nanopore sequencing has the potential ability to capture non-reference MEIs in complex genomic regions. Indeed, we observed 15 out of 17 nanopore-specific calls have reference *Alu*, L1, or LTR regions flanked by the insertion site. The assembly method or PacBio subread mapping (typically shorter than nanopore reads) could have difficulty in these reference repeat regions, while our targeted approach with nanopore sequencing could overcome the obstacles (with an average N50=22.6kb in the 17 flow cells of our experiments, **Supplementary Table 3**).

We have added a new **Supplementary Table 8** with the revised manuscript. We have revised the corresponding numbers, text, **Figure 4**, **Supplementary Fig.8-10**, **Supplementary Table 7,9**, Figure legends, and Supplementary information based on the updated non-reference MEI numbers that nanopore sequencing captured. We also added more text in the Results and Discussion section. Added and revised text in the manuscript is noted below.

Pg. 13 Cas9 enrichment and nanopore sequencing captures non-reference mobile elements in complex genomic regions: We further examined the set of MEIs that were captured exclusively by Cas9 targeted enrichment and nanopore sequencing, but not found in the PacBio-MEI intersection (**Supplementary Table 7**). We identified 12 additional L1Hs insertions as nanopore specific with ≥ 4 supporting reads that had been missed by the PacBio-MEI set with valid hallmarks, including target site duplication motifs, poly(A), EN Cleavage site, and empty site sequences, indicating a retrotransposition event induced by target-primed reverse transcription mechanism (TRPT) (**Supplementary Table 8**). In addition, we detected 5 *AluY* elements that were specifically captured by nanopore reads in the GM12878 genome (**Supplementary Table 8**). After refinement and inspection, we generated a full set of non-reference MEIs (194 L1Hs, 606 *AluY*, and 49 SVA) captured by Cas9 enrichment and nanopore sequencing in the GM12878 genome (**Supplementary Table 9**). Of note, all intermediate-value calls of PacBio-MEI for L1Hs, *AluYb*, and SVA_F were recovered in this study (**Supplementary Table 5,7,9**). Additionally, 46 calls with L1Hs sequence and 14 with *AluY* sequence were captured by Cas9 target nanopore sequencing yet not included in the final non-reference callset (**Supplementary Table 7**). Though lacking the support of TPRT hallmarks indicating a retrotransposition event, they may represent polymorphic duplicated sequences harboring an existing L1Hs or *AluY* element.

Pg. 14 Discussion: As we expected, in the nanopore-specific MEIs captured by our experiments, 15 out of 17 have reference *Alu*, L1, or LTR regions flanked by the insertion site. The assembly method or PacBio subread mapping (usually shorter than nanopore reads) could find difficulties in these reference repeat regions, where the Cas9 target method with nanopore sequencing could overcome the obstacles (with read length N50 ranged from 14.9Kbp to 32.3Kbp in the 17 flow cells of our experiments, **Supplementary Table 3**).

Pg. 24 Methods, Inspection and validation of nanopore-specific non-reference MEIs: A further in-depth inspection was employed for the non-reference calls categories as nanopore-specific. Each call has been inspected by two sections: a) general information including TRPT hallmarks (TSD motifs, poly(A), EN Cleavage site, and empty site sequence), length, strand, genotype, and population frequency in 32 genomes reported by Ebert et al. 2021, and b) IGV screenshot in a range of genomic region with genomic content annotation. The

TRPT hallmarks were identified by Cas9 target nanopore reads or from PacBio assembled contigs (**Supplementary Table 8**). A recurrence plot analysis was employed for further in-depth validation as well.

3) I felt the authors could perhaps have discussed more plainly the limitations of this study and method. MEIs have been mapped with ONT sequencing applied to human tissues elsewhere (Ewing et al, above) and this really needs to be accounted for correctly in several places in the text. That the MEI mapping done by Ewing et al is not mentioned at all is difficult to understand, at least from a scientific standpoint. Semantic claims of novelty in the Discussion are unnecessary as well. Also, as the authors bring up potential for mapping MEIs in neurodevelopment, they should cite the two studies reporting these with clear evidence (Evrony et al Neuron 2015; Sanchez-Luque et al Mol Cell 2019).

We thank the reviewer for pointing out where essential work was necessary to cite and apologize for their omission. As noted above in (1), we now provide a comparison of MEI read mappings between the projects. We have also now integrated Evrony et al Neuron 2015 and Sanchez-Luque et al Mol Cell 2019 into our discussion section when commenting on mosaic retrotransposition events.

Pg. 16 Discussion: Since Cas9 enrichment has been used to target rare rearrangement events (Stangl et al. 2020), implementation of this approach to detect unique, *de novo*, or somatic mobile element insertions is a feasible endeavor. This is emphasized by mounting evidence of MEIs escaping repression in embryogenesis, such as 5' truncated LINE-1s dodging repression by YY1 in neurons (Sanchez-Luque et al 2019). Nanopore-based identification of particularly rare insertion events, such as mosaic MEIs, is plausible. Recent work using whole genome sequencing of neuronal cells on nanopore platforms has demonstrated lineage tracing of retrotransposition events in early embryogenesis (Evrony et al. 2015). The application of targeted enrichment approaches to genetic mosaicism may improve the rate and depth at which this type of variation is sequenced.

4) More importantly, some discussion of technical limitations alongside merits would be helpful to readers. The necessary focus on L1HS guides at the 3' end (though there is some subfamily diversity towards the 5' end) leaves open two technical limitations: firstly, how often are reads spanning full-length L1HS MEIs recovered and, secondly, can L1HS promoter methylation be surveyed with this approach? Being able to survey methylation and other DNA modifications is a major advantage of ONT sequencing, and particularly Cas9 enrichment. This would have been useful to highlight even if an analysis was not performed.

We agree with the reviewer that additional technical limitations should be discussed in the manuscript. First, we have addressed some considerations in the enrichment levels and on-target reads in the discussion as a response to the reviewers first comment above. In addition, we agree with the reviewer that the ability to assess methylation is a large advantage to applying nanopore technology. As the reviewer points out, there is some limitation to guide selection, though as demonstrated in **Figure 2** there are suitable guides toward the 5' end of L1HS. However, in addition to L1HS specificity, we selected a guide near the 3' end with a bias toward sequencing toward the promoter to allow capture of partial L1HS insertions. However, the enrichment nanopore sequencing length prevents this from limiting our analysis of full-length inserts. We now provide an analysis of methylation patterns in full-length L1HS using the approach outlined in Ewing et al. and show consistent results with what was reported there as a new **Supplementary Fig.11**. We further report that, of the reads belonging to captured full-length L1HS, 65% extend across the consensus L1 sequence and beyond the L1HS promoter regions in the 3'->5' direction, indicating the ability to investigate L1HS promoter methylation in Cas9 targeted

nanopore sequencing experiments. This high capture rate is due to the strand bias in Cas9 enrichment (**Fig. 2**) and our high N50.

We have revised the Methods and Discussion, as well as provided scripts to reflect our methylation analysis in the Github site of this project.

Supplementary Fig.11: Examining CpG methylation of captured L1Hs reads.

a, L1Hs methylation profile over consensus L1 sequence in reference (blue) and non-reference elements (magenta). **b**, An example of methylation profile at chr3:151430755 non-reference L1Hs (black arrow 3'→5' orientation). Individual reads with methylation profiles are shown (top) along with their aggregate methylation profile (bottom).

Pg. 24 Methods. Analysis of L1Hs CpG Methylation: Nanopolish (Simpson et al. 2017) was used to call methylation on pooled GM12878 MinION and Flongle runs. Reference L1Hs methylation profiles were generated using methylartist (<https://github.com/adamewing/methylartist>), where methylation was aggregated across L1HS intervals from RepeatMasker (Smit et al. 2013). To generate profiles for non-reference elements, we built contigs using reads supporting full-length L1Hs and ±100kb of flanking sequences. Reads within 500bp of insertion sites were then extracted from the merged data and aligned to the constructed contigs using minimap2 (Li et al. 2018). These alignments and reads were then used for methylation calling with nanopolish.

The data from nanopolish was aggregated across L1Hs sequence in the constructed contigs and used for locus and consensus plotting using methylartist.

Pg. 15 Discussion: In addition, nanopore sequencing offers the ability to detect DNA modifications like 5mC. Even though we obtained nanopore reads with two directions, of the reads belonging to captured full-length L1Hs, 65% extend across the consensus L1 sequence and beyond the L1Hs promoter regions in the 3'->5' direction, indicating the ability to investigate L1Hs promoter methylation in Cas9 targeted nanopore sequencing experiments. We further examine CpG methylation profiles of full-length reference and non-reference MEIs (**Supplementary Fig.11**), showing consistent results with a prior study (Ewing et al. 2020).

Reviewer #2 (Remarks to the Author):

Review of McDonald et al, "Cas9 targeted enrichment of mobile elements using nanopore sequencing" Synopsis:

This well-written manuscript describes the application of Cas9-based long-read targeted sequencing of mobile element insertions. The work is unique in that the method employs hybridization-free and PCR-free sequencing of transposable elements. Characterization of such elements using long-reads provides many advantages over short-read based assembly approaches. For example, the distance between MEI elements using this approach can approach 100kb lengths due to the ultra-long read lengths of nanopore sequencers. Cleavage of the MEI only has to occur once and nanopore sequencing can initiate at that position, which is a key advantage of the technique.

In addition to the experimental work, the authors have developed a computational pipeline (Nano-Pal) to characterize known and novel elements, particularly elements that are embedded in complex genome regions. The research problem is interesting, and the authors chose a good model system with plenty of prior characterization (GM line, including trio) for method development.

The manuscript is technically sound overall. However, stronger validation of the novel MEIs using an orthogonal measurement (e.g., PCR-based targeting, with primers that flank the non-reference MEI) would strengthen the paper. The error-prone nature of single-pass nanopore sequencing brings up the question of the resolvability of closely related transposable element family members, given the challenge of deriving consensus sequences from ONT reads.

The authors present sufficient detail to reproduce the work. The statistical analyses and evidence was also adequate.

We thank Referee #2 for their appreciation of our work as well as for their thorough suggestions for improvement.

Major comments:

1. Figure 2a-c – Unclear on which plotting elements correspond to left versus right Y-axis. Why are the gRNAs for the L1Hs more evenly dispersed compared to Alus and SVAs?
2. Figure 2a-c – What is the reasoning for including the spatial distribution of gRNAs, when a single candidate was chosen? Is this to show additional potential targets along the span of the MEI for future work? Authors should indicate the position of the gRNA selected for this study (e.g., a red arrow).

We appreciate the comment #1 and comment #2 from the reviewer to help improve the graphical representation of our data.

To be noted, the length and substructure of different categories of MEIs vary a lot. The full-length LINE-1 element is ~6kb in length, containing a 5'UTR that harbors an internal RNA polymerase II promoter, two open reading frames (ORF1 and ORF2), and a 3'UTR that ends in a polyadenosine (poly(A)) tract. The full-length *AluY* element is 300bp long and contains the left and right monomers inside. The monomers were derived from 7SL RNA gene and thus they share a high level of sequence identity. *AluY* elements also have a A-rich connector between two monomers, and a poly (A) tail in the 3' end. SVA is a composite element consisting of a (CCCTCT)_n hexamer simple repeat region at the 5' end, an inverted *Alu*-like region, a variable number of tandem repeats (VNTR) region, a short interspersed element of retroviral origin (SINE-R) region, and a poly-A

tail. The length of the SVA element can range from 1.5kb to more than 3kb, mainly due to the various copy numbers of VNTR region. Therefore, you could observe that L1Hs gRNAs appear more evenly dispersed than *Alu*Ys and SVAs mainly because a) the size of L1Hs is larger than *Alu*Ys and SVAs, b) the repetitive nature of substructure (e.g. hexamer cap region and VNTR regions in SVAs) and the high identity of the substructure to the other elements of the genome (e.g. some part of monomers in *Alu*Y elements, the *Alu*-like region and SINE-R region in SVA elements) hindered the uniqueness of the sequence and led these parts unqualified as a guide RNA serving to capture specific subfamily.

The guide RNA design is a vital process in this project and one of the reasons that we gained such high on-target enrichment rates in the experiments. We want to share our selection strategy of guide RNA as much as possible and also show additional potential targets to the reader that they could make use of in the future projects. We listed the sequences of all guide RNA candidates for L1Hs, *Alu*Yb, *Alu*Ya, SVA_E, and SVA_F in the **Supplementary Table 1**, and their distributions, genomic frequencies and prioritized tiers we categorized in **Supplementary Fig.1,2**. The selected gRNA were yellow highlighted in the supplementary figures. To make it more clear to the readers, we now added red arrows where the selected guide was in Figure 2 according to the reviewer's constructive suggestion.

We appreciate the reviewer's helpful comments and have revised our figure, figure legend, and text accordingly. We added text describing which is the line and which is the histogram in the figure legend and added red arrows where the selected guide was in Figure 2. Revised text in the manuscript is noted below.

Figure 2: Guide RNA design for MEIs and guide RNA cleavage-site distribution: **a**, Distributions of candidate guide RNAs (left Y-axis and the histogram) in the L1Hs consensus sequence and structure information. The right Y-axis and the line indicate frequency of corresponding candidates in the reference genome sequence. **b**, Upper panel shows the distribution for *Alu*Yb and the lower panel for *Alu*Ya. **c**, Upper panel shows the distribution for SVA_F and the lower panel for SVA_E. Red arrows in **a**, **b**, and **c** indicate where the selected guide is. **d**, Cleavage-site distribution of all guide RNAs in this project. The x-axis indicates the position where the read ends or begins, with the number depicting the base distance from the PAM site (NGG). The PAM site (NGG) is colored blue and guide RNA bases are highlighted by a rectangle. Bases outside of the guide RNA or the PAM site are colored grey. The y-axis is the number of nanopore reads counted. The upper bar represents reads with forward strand sequencing outward from the 3' end of the guide RNA (red arrow) and the lower bar represents reads with reverse strand sequencing outward from the 5' end of the guide RNA (purple arrow).

3. p3 – Suggest a statement in introduction that Cas9 targeting approaches can be combined with any long-read RNA-seq platform (ONT, PacBio).

We thank the reviewer for the suggestion. We have addressed this in the reviewer's point #7 below.

4. Introduction – It would be helpful if the authors could cite more past work that involve targeted sequencing strategies of repetitive and transposable elements and delineate the distinguishing features and advantages of their approach. Such points could also be made in the Discussion.

We appreciate the comment from Review #2. We now extend the introduction for different MEI capture methods and cite more papers for the readers.

Pg. 2 Introduction: Experimental approaches from paired-end fosmid sequencing (Beck et al. 2010; Kidd et al. 2010) to PCR capture-based approaches (Beck et al. 2011; Iskow et al. 2010; Faulkner and Garcia-Perez 2017; Ovchinnikov, Troxel, and Swergold 2001; Huang et al. 2010; Badge, Alisch, and Moran 2003) have been developed to capture MEIs, but they have the disadvantage of being low throughput. More recent methods (Erwin et al. 2016; Zhou et al. 2020; Flasch et al. 2019; Ha, Loh, and Xing 2016) combining short-read sequencing and MEI 3' end capture techniques provide a more reliable way for the MEI discovery. Such approaches can be used in the investigation of single-cell MEI profiles (Erwin et al. 2016; Zhou et al. 2020), yet they too are hindered by the aforementioned disadvantages due to the short-read dependence.

5. Discussion, page 13 – GM is well characterized, and the utility of the method is demonstrated by the discovery of 21 non-reference MEIs. Can the authors give an estimate for the number of non-reference MEIs likely discovered for patient-specific genomes? Or common on the potential of the technique? How many non-reference MEIs would be typically found in an uncharacterized genome?

The reviewer brings up an excellent point and this is a key utility of the method that we present. Our previous work has shown that anywhere upward of 50% of MEIs are missed (in particular L1Hs). Considering the approach that we present here is near saturation, we would expect to capture those additional elements in patient-specific genomes.

Pg. 15 Discussion: Although this is only a handful of previously overlooked insertions, it is a surprising result from such an extensively sequenced genome as GM12878. As we have shown previously, upwards of 50% of MEIs are overlooked in data generated from short read sequencing approaches²⁹. Taken together with our results showing near complete saturation for MEIs in a single enrichment experiment, we anticipate that this may be highly effective at mapping patient samples, where a substantially larger proportion of MEI sites may differ from the GM12878 reference.

6. , Discussionpage 13 – Can the authors comment on the properties of the 21 non-reference MEIs that caused them to be missed by previous approaches?

This is a pertinent question that was also brought up by Reviewer #1. As we comment above, in this project we are using the 'PacBio-MEI' callset as a benchmark to our experiments, which is a union set of the mapping-based PALMER callset and the assembly-based PAV callset. After the intersection of 'PacBio-MEI' and Cas9 nanopore sequencing callset, we manually inspected all 379 potential nanopore-specific non-reference MEIs in this project (**Supplementary Table 7**). In response to this comment, we revisited these

21 non-reference MEIs that passed the first run of inspection and rigorously checked their hallmarks, including TSD motifs, poly(A), EN Cleavage site, empty site sequences, which indicate real a retrotransposition event induced by target-primed reverse transcription mechanism (TRPT). We also inspected their general information with IGV screenshot and genomic information annotated. This information and analysis can be found in the new **Supplementary Table 8**. The holistic application of these signals allowed us to confirm 12 non-reference L1Hs and 5 non-reference *AluY* as true positive retrotransposition events. The other 4 non-reference L1Hs calls were not able to be conclusively determined as a non-reference MEI insertion.

In our previous publication Zhou et al. 2020 NAR, we developed PALMER to resolve non-reference MEIs through PacBio subreads. We set a stringent cutoff to finalize a non-reference L1Hs callset for NA12878. We published PAV in Ebert et al. 2021. Science, which is an assembly-based caller and provides a fine set of non-reference MEIs with the fully resolved sequences from assembled contigs, assuming that the assembly process was working well. However, it is possible that some calls could be missed by 'PacBio-MEI' possibly due to the bias from PacBio technology or whole-genome sequencing, or perhaps more likely the requirement of very long reads to accurately map these elements. Therefore, it is not surprising that we found several more MEIs that were missed by PacBio sequencing since the Cas9 target nanopore sequencing for MEI capture would make a compensation for the 'PacBio-MEI' callset. Furthermore, we observed 15 out of 17 nanopore-specific calls have reference *Alu*, L1, or LTR regions flanked by the insertion site. These reference repetitive sequences could affect the performance of the assembly method or PacBio subread mapping (usually shorter than nanopore reads), where the Cas9 target method with ultra-long nanopore sequencing reads could overcome the obstacles.

We have added a new **Supplementary Table 8** with the revised manuscript for the detailed information of 17 non-reference nanopore-specific MEIs. We have revised the corresponding text, Figures, and supplementary materials. Added and revised text in the manuscript is noted below.

Pg. 13 Cas9 enrichment and nanopore sequencing captures non-reference mobile elements in complex genomic regions: We further examined the set of MEIs that were captured exclusively by Cas9 targeted enrichment and nanopore sequencing, but not found in the PacBio-MEI intersection (**Supplementary Table 7**). We identified 12 additional L1Hs insertions as nanopore specific with ≥ 4 supporting reads that had been missed by the PacBio-MEI set with valid hallmarks, including target site duplication motifs, poly(A), EN Cleavage site, and empty site sequences, indicating a retrotransposition event induced by target-primed reverse transcription mechanism (TRPT) (**Supplementary Table 8**). In addition, we detected 5 *AluY* elements that were specifically captured by nanopore reads in the GM12878 genome (**Supplementary Table 8**). After refinement and inspection, we generated a full set of non-reference MEIs (194 L1Hs, 606 *AluY*, and 49 SVA) captured by Cas9 enrichment and nanopore sequencing in the GM12878 genome (**Supplementary Table 9**). Of note, all intermediate-value calls of PacBio-MEI for L1Hs, *AluYb*, and SVA_F were recovered in this study (**Supplementary Table 5,7,9**). Additionally, 46 calls with L1Hs sequence and 14 with *AluY* sequence were captured by Cas9 target nanopore sequencing yet not included in the final non-reference callset (**Supplementary Table 7**). Though lacking the support of TPRT hallmarks indicating a retrotransposition event, they may represent polymorphic duplicated sequences harboring an existing L1Hs or *AluY* element.

Pg. 14 Discussion: As we expected, in the nanopore-specific MEIs captured by our experiments, 15 out of 17 have reference *Alu*, L1, or LTR regions flanked by the insertion site. The assembly method or PacBio subread mapping (usually shorter than nanopore reads) could find difficulties in these reference repeat regions, where the Cas9 target method with nanopore sequencing could overcome the obstacles (with read length N50 ranged from 14.9Kbp to 32.3Kbp in the 17 flow cells of our experiments, **Supplementary Table 3**).

Pg. 24 Methods. Inspection and validation of nanopore-specific non-reference MEIs: A further in-depth inspection was employed for the non-reference calls categories as nanopore-specific. Each call has been inspected by two sections: a) general information including TRPT hallmarks (TSD motifs, poly(A), EN Cleavage site, and empty site sequence), length, strand, genotype, and population frequency in 32 genomes reported by Ebert et al. ⁴⁶, and b) IGV screenshot in a range of genomic region with genomic content annotation. The TRPT hallmarks were identified by Cas9 target nanopore reads or from PacBio assembled contigs (**Supplementary Table 8**). A recurrence plot analysis was employed for further in-depth validation as well.

7. Discussion – Based on the mechanism of the Cas9 enrichment, it seems like this technique could also be integrated with PacBio long-read sequencing, with minor modifications to the adapter ligation and development of a computational pipeline to process the data. The authors should mention this possibility in the introduction and discussion.

We appreciate that the reviewer has pointed this out. The upstream steps of the Cas9 enrichment technique (Gilpatrick et al Nature biotech 2020) are certainly flexible enough to substitute Nanopore for PacBio. In fact, Tsai and colleagues submitted a preprint (<https://doi.org/10.1101/203919>) on an approach leveraging Cas9 to enrich for disease associated short tandem repeat loci for PacBio sequencing. While it is plausible to leverage PacBio sequencing with Cas9 enrichment, and we may speculate the utilization of this for capture of mobile elements, we do not feel that it is appropriate to explore this in the scope of our paper. This work relies on some PacBio sequencing data, however the entirety of the experimental work was done using Nanopore sequencers. While it may be worth citing some recent literature pairing Cas9 enrichment and PacBio, speculation on the specific modifications to the protocol without authority or experimental data feels irresponsible and too distracting from the main topic of our work.

8. Discussion – The recent ONT and PacBio systems (e.g., PromethION, Sequel II) are also much higher in throughput (e.g., 4 million reads + in a single flow cell). This should be mentioned in the discussion with an estimate of the impact of 10X higher throughput on targeted MEI analysis. This may become important for increasingly rare targets or scaling up the method.

We thank the reviewer for their comments. Newer sequencing platforms designed for whole genome analysis would certainly provide additional on-target sequences for our approach. However, our goal was to utilize a high-throughput, low-cost platform for an efficient interrogation of MEIs. Based on our saturation analysis, we do not expect to benefit greatly from upgrading to higher throughput and higher cost platforms from Nanopore. For our purposes, performing additional enrichment experiments for MEIs would likely provide the reads required to confidently map/call insertions from all of the MEI families studied. However, it is important to note that for extremely rare events, scaling up the sequencing power may be critical for efficient detection. We have made a note of this in the discussion.

Pg. 16 Discussion: However, efficiently detecting genetic mosaicism through Cas9 enrichment may require larger sequencing platforms or methodological innovations, as the rarity of target sites in a sample increases.

9. Discussion, page 14 – The authors should outline the remaining limitations/challenges of the approach.

In particular, discussion about the reasons for low target enrichment for certain families based on guide specificity, frequency, or any other factors, would be informative to the reader.

We now included new text in the discussion explaining our interpretation of these differences. We address this more specifically in response to the reviewer in response #9 below including the new text included in the manuscript.

What are the possibilities for scale-up of this approach and potential use in the clinic? Where are the limitations in terms of resources (e.g., library prep, sequencing costs)?

We do believe that this could be directly applied for clinical interpretation of mobile element insertions, though this is generally beyond the scope of this manuscript. We have added an analysis of targeted enrichment of MEIs compared to much more expensive whole genome nanopore sequencing above in response #1 to reviewer 1.

Though error correction of the reads were done (e.g., CANU), ONT reads are still error prone (e.g., <95% accuracy). Are there closely related MEI families for which there may be more difficulty resolving the subfamilies based on high sequence homology?

It is possible that certain subfamilies of MEIs with high sequence homology will be difficult to distinguish from each other. Increased sequence coverage enhances our ability to assign the sequenced element despite the error prone nature of nanopore sequencing. However, clinical interpretation of a pathogenic mobile element insertion event is likely to benefit more from the location of the insertion (which we can easily determine) rather than the specific subfamily of the event.

Minor comments:

1. p2 – define L1 (as acronym for LINE)

We describe L1 as defined in the field as a sub-group of the greater LINE family. We have now included “LINE-1” as an alternative representation that is also often used.

Pg. 2 Introduction: L1 (or, LINE-1) represents a subclass of LINEs and L1-derived sequences comprise approximately 17% of the human genome.

2. p2 – SVA acronym needed

We have now included the acronym for SVA:

Pg. 2 Introduction: SVA (SINE-VNTR-*Alu*) elements are active chimeric elements that have recently evolved and are derived from a SINE-R sequence coupled with a VNTR (variable number of tandem repeats) region and an *Alu*-like sequence.

3. Figure 1b – I suggest a split in the Time vs Current plot, because as is, it looks like the sequencing is done contiguously for the left/right MEI sequences, which is not the case.

We have clarified Figure 1b by splitting the current vs time plot to better illustrate the bidirectional sequencing shown above.

4. Figure 1c – Suggest specifically defining “MEI signal”. “The yellow bar represents MEI consensus sequence or MEI signals in pairwise comparison of Nano-Pal.” is unclear. I interpret it as the dominant portion (rather than the shorter portion) of the MEI post-Cas9 cleavage. Is there a significance to the MEI element being shorter in “MEI signal on one end”?

The MEI signal can be either the short or long fragment of the MEI post-Cas9 cleavage as described in the manuscript. There is no significance to the different lengths of the yellow bars. We have clarified the figure by making the yellow MEI signal bar lengths all the same size.

5. p3-4 – Please define Nano-Pal either at the end of the introduction or beginning of the results. Nano-Pal is described in Figure 1 without previous description that it is the computational pipeline for analysis of Cas9-enriched ONT data. Some aspects of the introduction to Nano-Pal on page 6 can be moved up for clarity.

We thank the comment of Referee #2. We have added text to describe the Nano-Pal in the introduction part.

Pg. 3 Introduction: By targeting the Cas9 to subfamily-specific sequences within each element and developing a computational pipeline (Nano-Pal) for analysis of Cas9-enriched nanopore sequencing data, we demonstrate enrichment of mobile elements across the genome that are both annotated and unannotated in the GRCh38 reference (reference and non-reference MEIs, respectively).

6. Supp Figure 3 – What is “maize bar”?

Maize is a shade of yellow and, along with blue, is one of the school colors of the University of Michigan. For clarification though, we have changed this to ‘yellow’.

7. Supp Figure 4 legend –“off-targeted” to “off-target”

We appreciate the correction from the reviewer and fixed the issue.

8. p6, paragraph 2 – What is the significance of the lower lengths for reads with MEIs on both ends? Is this expected based on the spatial distribution of L1Hs across the genome, compared to Alu and SVA elements?

Reads that have a single MEI on one end result from Cas9 making a single cut. By the nature of the cut, the strand will sequence continuously until either the fragment ends or the pore fails. Alternatively, reads with MEIs on both ends result from Cas9 excising a fragment of DNA by cutting at each MEI. These will inherently be shorter as they are cut from a larger, continuous DNA fragment. The number of the candidate targets in the genome, as well as their spatial distribution, will likely influence the size disparities between reads with a single MEI and reads flanked by MEIs.

9. What is the reason for the low on-target rates for the SVA family MEIs? Perhaps it would be helpful to add another column to compare the frequency in the genome for each MEI as a function of on-target rates.

We suspect that there are at least a few separate factors that influence the enrichment rates observed. One factor is likely the sheer number of targetable SVAs (SVA_F and SVA_E) based on the conditions of our experiments. Our **Supplementary Table 5** estimates the discrete number targetable MEIs for each sub family based on guide RNA mismatch thresholds. In the intermediate category (where the guide sequence may mismatch to the target by up to 3 nucleotides and what we deem most reasonable for Cas9 chemistry), our estimations indicate that there are several fold more L1Hs than either SVA subfamily. Similarly, the high number of *Alus* in the genome increases the set of near-matched guides and so we have a high background of other *Alus* enriched.

In addition, while we implemented a pipeline to prioritize guide RNAs, this was entirely *in silico* and we did not perform any comparative functional assays to evaluate guide RNA activity. It is possible that even among the Tier 0 guide candidates there are measurable performance differences that would influence the efficiency of targeting.

Pg. 14 Discussion: However, we observe that some of our MEIs targets (SVAs and *Alus*) have reduced enrichment compared to L1Hs. While we find relatively stable on-target rates for a specific guide target, differences in guide target rates can be expected due to different numbers of genomic integration sites, high numbers of similar target sequences, or guide RNA efficiency variation. For example, the low number of SVAs in the genome resulted in rather low on-target reads, but sufficient coverage to identify most of these elements in the genome. Similarly, the high number of *Alus* in the genome increases the set of near-matched guides and so we have a high background of other *Alus* enriched.

10. Supp Figure 5 – Define PAV

We describe the definition of PAV in the Methods and now added additional details in the legend of **Supplementary Fig.6** (original **Supplementary Fig.5**).

Supplementary Fig.6: Venn diagram of the PALMER callset and the PAV callset for non-reference MEIs in NA12878 genomes: PALMER callset (red circle) is from PacBio raw sub-reads, and PAV ([the Phased Assembly Variant caller](https://github.com/EichlerLab/pav), <https://github.com/EichlerLab/pav>, see Methods) callset (blue circle) is from PacBio assembly-based pipeline. The circles are depicted by the scale of the numbers showed inside. The union of two sets is generated as ‘PacBio-MEI’ to be the gold standard set to compare with the calls from nanopore data.

11. Discussion – Do the authors have an open source tool for design of gRNAs?

Currently, we don't have an open source tool for the design of gRNAs for mobile elements. However, we now provide the command lines/pipelines we used for the design of gRNAs, which is available at the Github site of this project:

<https://github.com/Boyle-Lab/NanoPal-and-Cas9-targeted-enrichment-pipelines>

12. p15 – “Two volumes of 100% ethanol...”

We have replaced ‘2’ with ‘Two’ at the beginning of this sentence.

13. p15 – “~250-500uL”, uL to microliters or mew symbol

We have replaced all uL with μ L.

14. p20 – “(MAPQ<10)”

We have fixed this spacing issue.

Reviewers' Comments:

Reviewer #1:

Remarks to the Author:

The authors have fully addressed my criticisms. I appreciate the way they went about this - well done.

I have just one remaining minor point: during revision, the authors added the following text to the Discussion: "Recent work using whole genome sequencing of neuronal cells on nanopore platforms has demonstrated lineage tracing of retrotransposition events in early embryogenesis (Evrony et al. 2015)." It would be best to remove "on nanopore platforms" from this sentence as that could lead to confusion (study was Illumina scWGS).

Reviewer #2:

Remarks to the Author:

The authors have addressed all the issues raised in the original review.

Reviewer #1 (Remarks to the Author):

The authors have fully addressed my criticisms. I appreciate the way they went about this - well done.

I have just one remaining minor point: during revision, the authors added the following text to the Discussion: "Recent work using whole genome sequencing of neuronal cells on nanopore platforms has demonstrated lineage tracing of retrotransposition events in early embryogenesis (Evrony et al. 2015)." It would be best to remove "on nanopore platforms" from this sentence as that could lead to confusion (study was Illumina scWGS).

Reviewer #2 (Remarks to the Author):

The authors have addressed all the issues raised in the original review.

We thank the reviewers for their positive reviews. We have removed "on nanopore platforms" as reviewer 1 suggests.